# RODS: Robust Optimization Inspired Diffusion Sampling for Detecting and Reducing Hallucination in Generative Models

**Yiqi Tian** [* † ‡]
ytian0@mgh.harvard.edu
yit30@pitt.edu

**Pengfei Jin** [* †]
pjin1@mgh.harvard.edu

**Mingze Yuan** [† §]
mingzeyuan@g.harvard.edu

**Na Li** [§]
nali@seas.harvard.edu

**Bo Zeng** [‡ ¶]
bzeng@pitt.edu

**Quanzheng Li** [† ¶]
li.quanzheng@mgh.harvard.edu

## Abstract

Diffusion models have achieved state-of-the-art performance in generative modeling, yet their sampling procedures remain vulnerable to hallucinations—often stemming from inaccuracies in score approximation. In this work, we reinterpret diffusion sampling through the lens of optimization and introduce RODS (Robust Optimization–inspired Diffusion Sampler), a novel method that detects and corrects high-risk sampling steps using geometric cues from the loss landscape. RODS enforces smoother sampling trajectories and *adaptively* adjusts perturbations, reducing hallucinations without retraining and at minimal additional inference cost. Experiments on AFHQv2, FFHQ, and 11k-hands demonstrate that RODS maintains comparable image quality and preserves generation diversity. More importantly, it improves both sampling fidelity and robustness, detecting over 70% of hallucinated samples and correcting more than 25%, all while avoiding the introduction of new artifacts. We release our code at https://github.com/Yiqi-Verna-Tian/RODS.

## 1 Introduction

Diffusion models [19, 25], also known as score-based generative models, have achieved remarkable success across various domains such as image generation [16, 43], audio synthesis [29, 10], and video production [20, 48]. These models generate data through an iterative denoising process that progressively transforms noise input structured outputs, offering a flexible trade-off between computational cost and sample quality [19, 25]. Beyond their generative capabilities, diffusion models have proven effective in solving complex inverse problems such as image inpainting [34, 36], colorization [45, 31], and medical imaging applications [13, 40]. This wide range of applications underscores the significant potential of diffusion models to transform different fields by generating high-quality, diverse synthetic data that can be tailored for specific needs.

Despite their impressive performance, diffusion models are prone to hallucinations—outputs that are unfounded, unfaithful, or not grounded in the underlying data [24, 30]. While extensively studied in text [41, 14], hallucination is equally problematic in visual domains. For instance, diffusion models

---

[*]Equal contribution.

[†]Center for Advanced Medical Computing and Analysis, Massachusetts General Hospital and Harvard Medical School, Boston, MA 02114

[‡]Department of Industrial Engineering, University of Pittsburgh, Pittsburgh, PA 15261

[§]School of Engineering and Applied Sciences, Harvard University, Boston, MA 02138

[¶]Corresponding author.

39th Conference on Neural Information Processing Systems (NeurIPS 2025).

may synthesize anatomically implausible humans [2, 38] or generate unrealistic scene compositions. In high-stakes applications such as medical imaging or radar sensing, these errors can lead to false diagnoses or misinformed decisions [27, 11]. A key difficulty is that hallucination lacks a formal definition, making both detection and mitigation inherently challenging [24]. Nonetheless, its importance has sparked growing research to understand and address these failures.

Recent mitigation strategies in diffusion models include trajectory variance filtering, which detects off-manifold samples via prediction instability in the denoising path [2]; local diffusion techniques that isolate out-of-distribution regions and process them separately [27]; and early stopping methods that truncate sampling before overfitting to noise occurs [51]. Additionally, fine-tuning methods have also shown promise, particularly in hallucination-prone areas like hands and faces, using structure-aware losses [32] or subject-specific adaptation [44]. While these methods address specific aspects of the problem, hallucination mitigation remains an open challenge. Existing approaches vary in scope, assumptions, and evaluation criteria, and the field still lacks standardized benchmarks for systematic comparison.

To understand how diffusion samplers can fail, it is helpful to view sampling as an optimization process. Consider the probability flow ordinary differential equation (PF-ODE) as guiding a climber down a mountain ridge in dense fog. At each time $t$, the learned score function provides a direction of descent; as time progresses ($t \downarrow 0$), the fog lifts and the terrain becomes clearer. This step-by-step refinement closely resembles the continuation method in numerical optimization, where one solves a sequence of gradually harder problems locally to approach a final objective. However, when the score function is imperfect—as is often the practice case—it can point in misleading directions. In benign regions, such errors are tolerable, but in sensitive areas, they may flip the intended descent direction entirely, sending the trajectory off-manifold and leading to hallucinated outputs [46, 4, 2, 39].

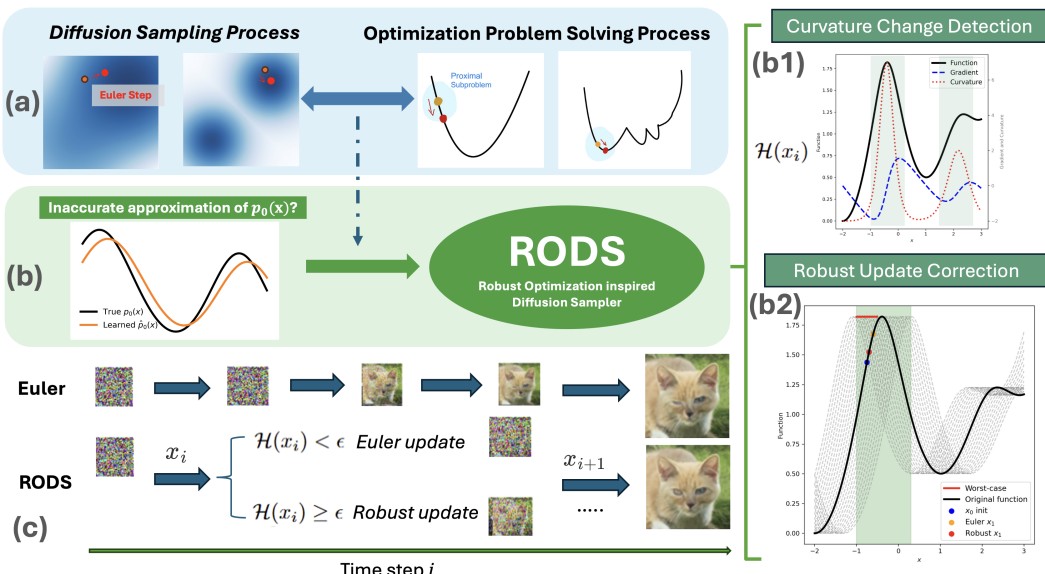

Figure 1: The roadmap of our paper: (a) Section 3 formulates the diffusion sampling process as an optimization problem solved based on the continuation method. (b) Section 4 introduces the RODS framework to address the inaccurate approximation of the score function: (b1) Section 4.1 details how we detect high-risk regions based on local curvature changes (highlighted in green). (b2) Section 4.2 describes how robust updates are performed to mitigate potential hallucinations. (c) Section 5 presents experimental results and analysis, illustrating improvements in hallucination detection and correction.

This observation motivates a simple yet powerful idea: sample as a cautious optimizer. Instead of blindly following the indicated direction, probe the surrounding neighborhood for signs of instability, such as regions with rapidly changing curvature. This perspective naturally leads to the principle of robust optimization (RO)—a strategy that accounts for worst-case perturbations to ensure progress even under uncertainty. We operationalize this idea through a plug-and-play method called the Robust

Optimization–inspired Diffusion Sampler (RODS). At each step, RODS: (i) explores a small local neighborhood of radius $\rho$ to detect directions where the score landscape exhibits high curvature or instability, and (ii) adjusts the update direction to maintain objective descent under worst-case local perturbations. Importantly, RODS requires no retraining of the diffusion model and introduces minimal computational overhead.

The roadmap of our paper is summarized in Figure 1. The key contributions of this work are as follows:

- We reinterpret diffusion sampling through the lens of continuation methods in numerical optimization, which frames each sampling step as a progressively refined sub-problem and opens the door for leveraging various optimization tools.

- We propose RODS, a plug-and-play sampling framework that integrates RO principles into the diffusion sampling process. RODS includes a novel curvature-based change detection step that probes local geometry to identify sudden variations in score field behavior. While not detecting hallucinations directly, these changes often signal high-risk regions that correlate with failure modes such as hallucination. According to the detection of the high-risk regions, RODS then dynamically adjusts sampling directions based on worst-case local perturbations, enhancing robustness and reliability in uncertain or error-prone regions.

- We conduct extensive numerical experiments on high-dimensional, real-world image datasets—including AFHQv2, FFHQ, and a Hand dataset—that go beyond synthetic 1D or 2D Gaussian examples. Our results show that while maintaining comparable image quality and generation diversity, RODS successfully detects over 70% of hallucinated samples and corrects more than 25%, all without introducing new artifacts or requiring any modification to the pre-trained diffusion model.

## 2 Preliminary

### 2.1 Diffusion Model

Diffusion models are probabilistic generative models that transform data into noise via a forward stochastic process and then reconstruct it by reversing that process. Formally, the forward process is defined by the stochastic differential equation (SDE):

$$dx_t = \mu(x_t, t)\, dt + \sigma(t)\, dw_t, \tag{1}$$

where $w_t$ is standard Brownian motion and $t \in [0, T]$. Under mild conditions, this SDE admits a deterministic counterpart—the probability flow ODE [49]:

$$dx_t = \left[ \mu(x_t, t) - \frac{1}{2}\sigma(t)^2 \nabla \log p_t(x_t) \right] dt, \tag{2}$$

which preserves the same marginal distributions $p_t(x)$.

In EDM method [25], the forward process is defined by zero drift, $\mu(x_t, t) = 0$ and variance $\sigma(t) = \sqrt{2t}$, resulting in Gaussian smoothing $p_t = p_0 * \mathcal{N}(0, t^2 I)$, where $*$ denotes the convolution operator. To generate samples, one trains a neural network to approximate the score function $s_\theta(x, t) \approx \nabla \log p_t(x)$, and substitutes it into the reverse ODE. This yields different parameterizations that are mathematically equivalent but tailored to different training objectives and architectural designs:

$$\frac{dx_t}{dt} = -t s_\theta(x_t, t) = -\epsilon_\theta(x_t, t) = \frac{x_t - D_\theta(x_t; t)}{t}, \tag{3}$$

where $s_\theta$, $\epsilon_\theta$, and $D_\theta$ denote the score, noise prediction, and denoising function, respectively, as used in different methods. Starting from a Gaussian distribution at large $t$, one iteratively removes noise to recover samples from $p_0$. This framework has shown strong empirical results across image, audio, and other generative tasks.

### 2.2 Continuation Method

The continuation method [3] is a powerful optimization technique designed to handle challenging, nonconvex problems by progressively transforming the optimization landscape. It begins by

introducing a regularization term $\lambda R(x)$ to the objective function:

$$\min_x f(x) + \lambda R(x), \tag{4}$$

where $f(x)$ is the original objective function, and $R(x)$ imposes a prior or penalty on the solution. The parameter $\lambda$ controls the influence of the regularization term.

The key idea is to simplify the problem early in the optimization process by setting $\lambda$ to a large value, allowing the regularization term to dominate and smooth out the optimization landscape. As the optimization proceeds, $\lambda$ is gradually reduced, making the problem increasingly faithful to the original objective $f(x)$. At each step, the solution to the simpler problem is used as the starting point for the next iteration. This iterative reduction in $\lambda$ continues until it approaches zero, at which point the algorithm converges to a solution of the original problem. See Appendix A.1 for algorithm details. This method is particularly well-suited for nonconvex problems, where poor local minima can hinder direct optimization approaches. This method balances tractability and accuracy, smoothing out local irregularities in the optimization landscape while progressively honing in on the true objective.

### 2.3 Robust Optimization (RO)

Robust optimization (RO) [5] [6] [7] [33] is a fundamental framework in optimization that seeks solutions which remain effective under arbitrary perturbations with a defined set. In its standard formulation, RO is expressed as:

$$\min_{x \in \mathcal{X}} \max_{u \in \mathcal{U}} f(x, u),$$

where $\mathcal{U}$ denotes an uncertainty set that is defined to encapsulate potential variations or disturbances. Rather than optimizing a fixed objective, RO explicitly accounts for randomness within $\mathcal{U}$, thereby ensuring that the solution is resilient to adverse shifts in the problem data. This approach is particularly valuable in situations where model parameters or environmental conditions are uncertain, as it helps prevent performance degradation when faced with unexpected changes.

## 3 An Optimization View of Diffusion Sampling Process

Recent research has explored connections between diffusion models and optimization methods, broadly classified into two categories. The first category interprets Score-Matching Langevin Dynamics (SMLD) in diffusion models as variants of stochastic gradient descent (SGD) [49, 52, 50].The second category leverages diffusion models as learned priors for solving optimization tasks, such as planning or inverse problems [23, 12, 35]. Our work aligns more closely with the first category but distinctly frames diffusion sampling as a continuation or homotopy method from numerical optimization. Unlike conventional interpretations based on fixed-noise-level SMLD, we explicitly regard the entire diffusion trajectory as solving a sequence of progressively refined optimization subproblems. Further theoretical comparisons and detailed discussions are provided in Appendix D.

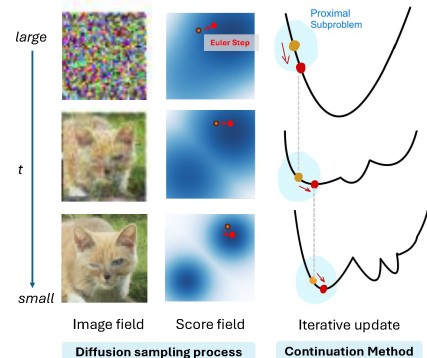

Figure 2: The equivalence between the diffusion sampling process and the optimization continuation method.

Our discussion so far treats diffusion sampling via an *ODE perspective*, writing

$$\frac{\mathrm{d}x}{\mathrm{d}t} = -\epsilon_\theta(x, t) \tag{5}$$

and integrating from time $t = T$ down to $t = 0$. Here, the neural network $\epsilon_\theta$ is trained such that

$$\nabla f_t(x) \propto -\epsilon_\theta(x, t), \quad \text{where } f_t(x) = -\log p_t(x).$$

Hence, each step of (5) resembles a *gradient descent* update on $f_t(x)$. In this section, we reinterpret the same steps as a *continuation method* in optimization, show that each Euler step can be viewed as solving a local subproblem via a *proximal* operator, and finally present a unified pseudocode.

**Analogy in Diffusion.** The continuation method aims to solve a challenging objective

$$\min_x \; f(x) + \lambda R(x)$$

by first introducing a strong regularization $R(x)$ with a large smoothing parameter $\lambda$, and then *decreasing* $\lambda$ gradually. One thereby transitions from a simpler (highly smoothed) problem to the original, more complex objective.

In diffusion models, the noise level $t$ plays the role of "smoothing strength." At a high noise $t$, the convolved distribution $p_t(x) = p_0(x) * \mathcal{N}(0, t^2 I)$ is more diffuse than $p_0(x)$, so its negative log-density $f_t(x) = -\log p_t(x)$ is relatively smooth and has fewer local minima than $f_0(x) = -\log p_0(x)$. As $t$ decreases, $f_t(x)$ transitions from a highly smoothed version to the original, sharper landscape. This parallels the continuation principle: one "continues" from an easy subproblem at high noise ($t \approx T$) to the more difficult, exact $-\log p_0(x)$ at $t = 0$. Such procedure equivalence is formalized in Theorem 1, with the proof detailed in the Appendix A.1.

**Theorem 1** (Procedure Equivalence). *Assume the image distribution can be approximated by a mixture of Gaussians, i.e.* $p_0(x) = \sum_{i=1}^K w_i \mathcal{N}(x \mid \mu_i, \sigma_0^2), \quad \sum_{i=1}^K w_i = 1.$ *Define*

$$f_t(x) = -\log p_0(x) - \log\Big\{\sum_{i=1}^K \tilde{w}_i(x) \, \exp\big(\tfrac{\|x - \mu_i\|^2 \, t^2}{2 \, \sigma_0^2 (\sigma_0^2 + t^2)}\big)\Big\},$$

*where* $\tilde{w}_i(x) = \frac{w_i \, \mathcal{N}(x \mid \mu_i, \sigma_0^2)}{\sum_{j=1}^K w_j \, \mathcal{N}(x \mid \mu_j, \sigma_0^2)}, \quad \sum_{i=1}^K \tilde{w}_i(x) = 1.$ *Then*

$$\min_x f_t(x) \quad\Longleftrightarrow\quad \min_x \big[-\log p_t(x)\big].$$

*Moreover,* $f_t(x)$ *shifts continuously from a heavily smoothed version at* $t = T$ *toward* $-\log p_0(x)$ *at* $t = 0$*, aligning with continuation.*

**Analogy in Euler Step.** Taking a closer look at the procedure, when discretizing the diffusion ODE, we move from $t_i$ to $t_{i-1} = t_i - \Delta t$ by a simple *Euler step*:

$$x_{t_{i-1}} = x_{t_i} - \Delta t \, \nabla f_{t_i}(x_{t_i}). \tag{6}$$

This update is precisely *one step* of gradient descent on $f_{t_i}(x)$, under a local trust-region interpretation where the step size $\Delta t$ controls how far we move from $x_{t_i}$. Below, we formally show that this *one-step gradient* can also be viewed as solving a local subproblem with a proximal (quadratic) penalty.

**Theorem 2** (Euler = One-Step Gradient = Proximal Update). *Define the local subproblem at time $t_i$:*

$$\min_y \Big\{ f_{t_i}(y) + \tfrac{1}{2\Delta t} \|y - x_{t_i}\|^2 \Big\}. \tag{7}$$

*A single gradient descent step on $f_{t_i}$ of size $\Delta t$ solves (7) approximately, with solution $y^*$ satisfying*

$$y^* = \mathrm{prox}_{\Delta t \, f_{t_i}}(x_{t_i}) \quad\Longleftrightarrow\quad y^* = x_{t_i} - \Delta t \, \nabla f_{t_i}(y^*),$$

*and if we approximate* $\nabla f_{t_i}(y^*) \approx \nabla f_{t_i}(x_{t_i})$*, this yields the Euler step (6).*

This shows that each Euler move *is* a single gradient descent update on $f_{t_i}(x)$, equivalently interpretable as a *proximal* (trust-region) solution at time $t_i$.

Overall, because diffusion lowers the noise level $t$ in discrete steps, we can view the entire sampling process as a *continuation method*: each subproblem corresponds to minimizing $f_{t_i}(x)$, starting from $t_N = T$ (high noise, smooth landscape) and ending at $t_0 = 0$ (true but sharp $-\log p_0(x)$). Algorithm 1 details this procedure. It is numerically identical to solving the ODE (5) with an Euler integrator but emphasizes the iterative optimization interpretation. Furthermore, the proximal operator on each subproblem (7) *matches* the Euler updates, as illustrated in Figure 2. More sophisticated ODE solvers (Heun, RK4) correspond to more elaborate ways of approximating $y^*$ in each subproblem.

**Algorithm 1** Diffusion Sampling as a Continuation-Method Optimization

**Require:**
- Time steps $\{t_i\}_{i=N}^0$ with $t_N = T$ and $t_0 = 0$,
- Step size $\Delta t$ (so $t_{i-1} = t_i - \Delta t$),
- Neural network $\epsilon_\theta(\cdot, \cdot)$ satisfying $\nabla f_t(x) = -\epsilon_\theta(x, t)/t$.

1: **Initialize:** $x_{t_N} \sim \mathcal{N}(0,\ t_N^2 I)$
2: **for** $i = N, \dots, 1$ **do**
3:      **Compute Gradient:**

$$\nabla f_{t_i}(x_{t_i}) = -\frac{\epsilon_\theta(x_{t_i}, t_i)}{t_i}$$

4:      **Proximal Subproblem:**

$$x_{t_{i-1}} = \text{prox}_{\Delta t \, f_{t_i}}(x_{t_i})$$

5:      **(Euler Equivalent):**

$$x_{t_{i-1}} = x_{t_i} - \Delta t \, \nabla f_{t_i}(x_{t_i})$$

6: **end for**
7: **Return:** $x_{t_0}$

---

**Algorithm 2** Robust Optimization–inspired Diffusion Sampler (RODS)

**Require:** Score network $D_\theta(x; \sigma)$, time steps $t_i \in \{0, \dots, N\}$, perturbation set $\mathcal{M}$, curvature threshold $\epsilon$
1: Sample $x_0 \sim \mathcal{N}(0, t_0^2 I)$
2: **for** $i = 0, \dots, N - 1$ **do**
3:      $d_i \leftarrow \frac{x_i - D_\theta(x_i; t_i)}{t_i}$
4:      $v(x_i) \leftarrow d_i/t_i$
5:      Compute curvature index:

$$\mathcal{H}(x_i) \leftarrow \max_{\delta \in \mathcal{M}} \|\nabla_{x_i}\|v(x_i + \delta)\| - \nabla_{x_i}\|v(x_i)\|\|$$

6:      **if** $\mathcal{H}(x_i) \geq \epsilon$ **then**
7:          $\delta_i \leftarrow \arg\max_{\delta \in \mathcal{M}} g_{t_i}(x_i, \delta)$
8:          $\hat{x}_i \leftarrow x_i + \delta_i$
9:          $\hat{d}_i \leftarrow \frac{\hat{x}_i - D_\theta(\hat{x}_i; t_i)}{t_i}$
10:         $x_{i+1} \leftarrow x_i + (t_{i+1} - t_i) \hat{d}_i$
11:     **else**
12:         $x_{i+1} \leftarrow x_i + (t_{i+1} - t_i) d_i$
13:     **end if**
14: **end for**
15: **return** $x_N$

---

# 4 Robust Optimization inspired Diffusion Sampler (RODS)

Recent studies have shown that diffusion models sometimes produce unrealistic or "hallucinated" outputs [38, 8], especially when sampling paths pass through low-density regions where the model's estimates become unreliable [2]. Standard sampling methods closely follow the model's estimated direction but lack built-in safeguards against sudden instabilities or sharp changes in behavior. Therefore, our proposed hallucination detection builds on the insight that hallucinated behavior is closely correlated with model uncertainty, which can be quantitatively inferred from the local curvature change (rapid fluctuations) of the score field $v(x) = \nabla_x \log P_t(x)$. For example, sharp increases in curvature proxy often signal transitions into unstable or poorly approximated zones, where hallucinations are more likely to occur.

From the *optimization perspective*, we could conduct a robust optimization idea to improve the reliability of sampling. Instead of directly minimizing the estimated potential function $f_t(x)$ at each timestep (i.e. each subproblem), we consider such function via a more cautious approach:

$$\min_x \ \max_{\hat{f}_t \in \mathcal{F}_t} \hat{f}_t(x),$$

where $\mathcal{F}_t$ represents a set of plausible variations around the estimated function. This setup asks: what is the worst-case outcome in the surrounding region, and how can we choose $x$ to avoid bad surprises? Intuitively speaking, by using RO method, we peek at the place where the model is most uncertain, borrow a direction that could most effectively mitigate potential errors, and use it to update the current one. As a result, we anticipate that, the sampling process becomes more stable, particularly in regions where the model's approximation is noisy or inconsistent.

## 4.1 Curvature Change Detection

To decide *when* RODS should switch from a standard to a robust update, we need a fast signal that a sampling step is entering unstable territory. As mentioned in [49], generalization errors arise during the diffusion reverse process in low-density regions. This results in the learned score function deviating from the sharp, high-gradient structure of the ground truth score, and leads to potential *hallucination effects* in the diffusion model, where generated samples may deviate from realistic

outputs. [2] proposed detecting such errors by measuring prediction instability along the *temporal direction* of the denoising trajectory. In contrast, our approach focuses on identifying instability in the *spatial domain*. High local curvature in the score field $v(x)$ serves as an indicator of model uncertainty and, consequently, hallucination risk. To detect landscapes of the score field, we propose using an index $\mathcal{H}(x)$, inspired by the maximum eigenvalue of the second derivative of the score norm:

$$\mathcal{H}(x) = \|\nabla_x\|v(x + \delta)\| - \nabla_x\|v(x)\|\|, \quad \delta = \arg\max_{\|\delta\|=\rho}\|v(x + \delta)\|, \tag{8}$$

where $\rho$ is the detection radius. Intuitively, $\mathcal{H}(x)$ measures the degree of fluctuation in the magnitude of the score along direction $\delta$. For theoretical insights, please refer to Appendix A.2. We consider the current step $x$ to have entered a high-risk region when $\mathcal{H}(x) > \epsilon$, where $\epsilon$ is a predefined detection threshold. In practice, the choice of $\epsilon$ reflects a trade-off: a lower threshold increases sensitivity and enables the model to detect more hallucinated samples, but may also lead to false positives by flagging benign regions as unstable. When the surrounding region is reliable (e.g., visually simple), a smaller $\rho$ is preferred, reflecting higher trust in local information. Conversely, a larger $\rho$ makes the approach more conservative, expanding the search when the neighborhood may be misleading.

More importantly, the threshold can be adapted in different applications. For example, in high-stakes domains such as medical imaging, or more challenge generation tasks, a lower $\epsilon$ may be preferred to prioritize sensitivity (Appendix B.2). Furthermore, all of our experiments show that applying updates to false positive samples does not degrade output quality at the cost of mild computation (as shown in the results section), indicating that RODS remains robust even in the presence of false positives.

## 4.2 Robust Sampling Process

Once we detect that the current sampling step lies in a high-risk region, we apply a robust update to make the diffusion process more stable. From an RO viewpoint, this means preparing for the worst-case scenario within a set of plausible perturbations to the function correspondingly. In this context, we introduce two robust update schemes grounded in the RO framework: *Sharpness-Aware Sampling (SAS)* and *Curvature-Aware Sampling (CAS)*. Both approaches operate by solving a min–max RO problem that seeks to guard against local failures, but they do so with different emphases.

**Sharpness-Aware Sampling (SAS).** SAS focuses on worst-case spikes in the score function $f_t$, which captures the model's energy landscape at time $t$ and can be directly derived from RO scheme. We define a function family $\mathcal{F}_t$ consisting of localized shifts of $f_t$:

$$\mathcal{F}_t = \left\{\hat{f}_{t,\delta} \,\middle|\, \hat{f}_{t,\delta}(x) = f_t(x + \delta), \, \delta \in \mathcal{M}\right\},$$

where $\mathcal{M}$ is a small bounded set (e.g., an $\ell_2$-ball) of perturbations. For the associated min–max problem, we minimize the worst-case value of $f_t$ within the nearby neighborhood, which hedges against sharp peaks or local inconsistencies that could derail the sampling trajectory. Specifically, we handle the following bilevel optimization problem:

$$\min_x \quad f_t(x + \delta), \quad \text{where} \quad \delta = \arg\max_{\delta' \in \mathcal{M}}\{\hat{f}_t(x, \delta') : \hat{f}_{t,\delta'} \in \mathcal{F}_t\}$$

Actually, as in the following, we can generalize the inner maximization to other forms

$$\delta = \arg\max_{\delta' \in \mathcal{M}} g_t(x, \delta'),$$

where $g_t(x, \delta')$ defines some type of (undesired) metric to focus on. The details of solving subproblems are provided in Appendix B.4.

**Curvature-Aware Sampling (CAS).** In contrast to SAS, which concentrates on worst-case increases in function values, CAS focuses on the shape of the score landscape. Specifically, it aims to identify directions where the slope of the function is steepest by maximizing the gradient norm:

$$g_{t,\text{CAS}}(x, \delta) = \|\nabla f_t(x + \delta)\|.$$

While this form differs from the classical RO objective, it shares the same spirit: guarding against dangerous regions. Large gradient magnitudes indicate steep or unstable directions, so stepping toward them helps uncover problematic curvature. After identifying the high-curvature direction, the sampler can take a controlled step to avoid abrupt changes.

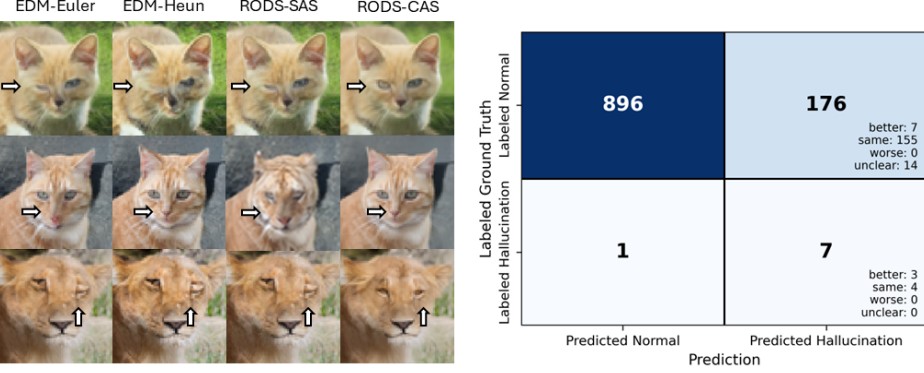

Figure 3: **AFHQv2** sampling results. **Left:** Visual comparison across different samplers—EDM (Euler, Heun), and our RODS-SAS, RODS-CAS. Hallucinations include misplaced eye, incorrect facial structures, etc. **Right:** Confusion matrix at $\epsilon = 0.1$ and $\max \|\delta\| = 1$ over 1,080 samples using the proposed hallucination index $\mathcal{H}(x)$. RODS-CAS detects 87.5% of labeled hallucinations, correcting 10 cases (better), and introduces no degraded cases (worse).

## 4.3 RODS Algorithm

Therefore, once we detect that the sampling path is entering a potentially unstable region, which typically characterized by sharp curvature changes, we apply a more cautious update to help prevent unrealistic or unstable samples. The full procedure is outlined in Algorithm 2. It closely follows the deterministic sampling style of EDM [25], but selectively activates robust steps when the local geometry suggests the model may be unreliable.

In each iteration, the algorithm first estimates the current score direction and checks whether the score landscape is changing too sharply using the curvature index $\mathcal{H}(x_i)$. If the region is flagged as high-risk, it proceeds through three steps: (1) Search – look around the current sample to find the most sensitive direction $\delta_i$ that exposes model uncertainty, obtained by maximizing a function $g_{t_i}(x_i, \delta)$, which can be tailored to different risk criteria (e.g., SAS or CAS); (2) Descent – compute the steepest descent direction $d_i$ at the identified worst-case point to neutralize local instability; and (3) Update – move the original sample $x_i$ along this safe corrective direction.

## 5 Experiment

We evaluate our method on three benchmarks: AFHQ-v2 [26], FFHQ [26], and 11k-hands [1]. For AFHQ and FFHQv2, we use pre-trained models from EDM [25]. For 11k-hands, we train a diffusion model using standard frameworks and configurations in EDM. All training and evaluation are conducted on a single NVIDIA A100 GPU (Appendix B.1 for dataset and training details).

Due to substantial differences across prior works in methodology, assumptions, and evaluation protocols, there is currently no standardized benchmark for hallucination detection, generation and correction. We summarize and contrast related methods in Appendix C. As a plug-and-play module, RODS can reduce to the default *EDM-Euler* sampler when the detection threshold $\epsilon$ is high. Therefore, *EDM-Euler* serves as our baseline method. In particular, we perform manual hallucination annotation for evaluation (Appendix B.6).

**AFHQv2** We generate 1,080 samples from the AFHQv2 dataset using random seeds from 0 to 119, with a batch size of 9. With the robust region $\mathcal{M} := \{\|\delta\| = 1\}$ and the detection threshold $\epsilon = 0.1$, our method detects 7 of 8 labeled hallucinations, improves 7 unlabeled cases, and introduces no degraded outputs—demonstrating its robustness. As shown in Figure 3, the left panel illustrates successful visual corrections by RODS-CAS, while the right panel summarizes detection and correction performance. Notably, most samples labeled as non-hallucinated by human annotators remain visually unchanged after the robust update (Appendix Figure 13 and 14). In some cases, however, even these non-hallucinated samples exhibit subtle improvements—appearing more semantically coherent after refinement (e.g., top-left in Appendix Figure 8).

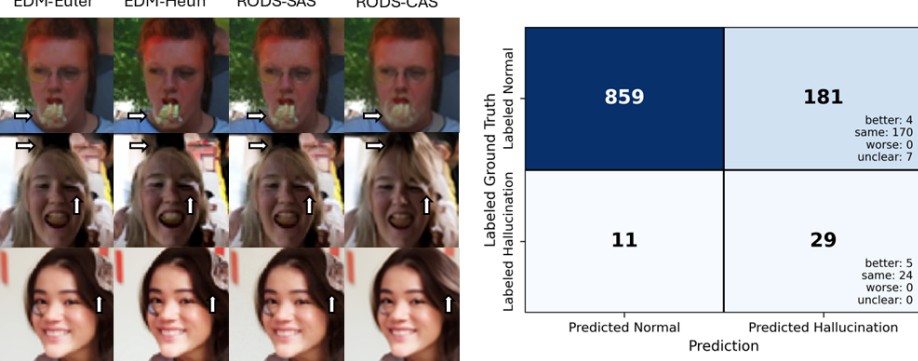

Figure 4: **FFHQ** sampling results. **Left:** Visual results across EDM-Euler, EDM-Heun, and our RODS-SAS, RODS-CAS. Hallucinations include distorted eye regions and implausible facial geometry. **Right:** Confusion matrix at $\epsilon = 0.09$ and $\max \|\delta\| = 8$ over 1,080 samples using the proposed hallucination index $\mathcal{H}(x)$. RODS-CAS detects 72.5% of labeled hallucinations, corrects 9 cases (better), and introduces no degradation (worse).

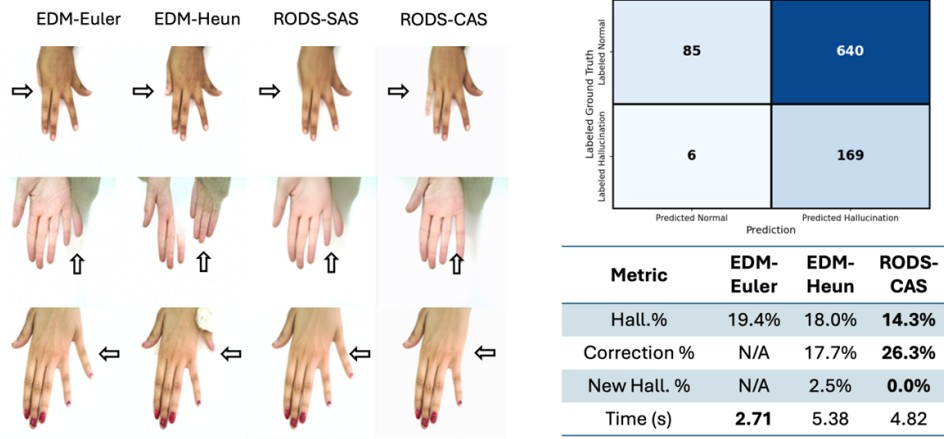

| Metric | EDM-Euler | EDM-Heun | RODS-CAS |
|---|---|---|---|
| Hall.% | 19.4% | 18.0% | **14.3%** |
| Correction % | N/A | 17.7% | **26.3%** |
| New Hall. % | N/A | 2.5% | **0.0%** |
| Time (s) | **2.71** | 5.38 | 4.82 |

Figure 5: **11k-hands** sampling results. **Left:** Visual results for different samplers. RODS-CAS corrects hallucinations such as extra or missing fingers while preserving anatomical plausibility. **Top-Right:** Confusion matrix at $\epsilon = 0.014$ and $\max \|\delta\| = 30$ over 900 samples using the proposed hallucination index $\mathcal{H}(x)$. **Bottom-Right:** Quantitative metrics: hallucination rate ($\downarrow$), correction rate ($\uparrow$), new hallucination rate ($\downarrow$), and inference time (in seconds).

**FFHQ** We then apply the same evaluation procedure to the AFHQ dataset, with $\mathcal{M} := \{\|\delta\| = 8\}$ and $\epsilon = 0.09$; an ablation study on threshold sensitivity is provided in Appendix B.2. Our method detects 29 of 40 labeled hallucinations and improves 9 of the detected images without introducing any visible worse cases. As shown in Figure 4, the left panel illustrates representative visual corrections, and the right panel reports quantitative results.

**11k-hands** To evaluate performance in a more challenging setting, we test on the 11k-hands dataset, where hallucinations are more frequent and severe. We generate 900 samples using seeds 0 to 99 with a batch size of 9. To ensure high sensitivity, we adopt a stricter configuration with $\mathcal{M} := \{\|\delta\| = 30\}$ and $\epsilon = 0.014$. Under this setting, our method successfully detects 169 out of 175 human-labeled hallucinations. As a trade-off for this high sensitivity, the method attempts to revise every suspicious sample.

Based on ablation studies, we found that the key issue lies in the *directional error*, particularly in low-density, high-uncertainty regions (Appendix B.5). Furthermore, we observe that critical steps (Appendix B.3), those most responsible for hallucinations, tend to occur in the middle of

Table 1: Comparison of FID, Inception-Var, and DreamSim metrics across datasets, each evaluated on 1k generated samples.

| Dataset | FFHQ | | | AFHQv2 | | | 11k-hands | | |
|---|---|---|---|---|---|---|---|---|---|
| | FID ↓ | Incep-Var ↑ | DreamSim ↑ | FID ↓ | Incep-Var ↑ | DreamSim ↑ | FID ↓ | Incep-Var ↑ | DreamSim ↑ |
| EDM-Euler | 20.48 | 0.0669 | 0.407 | 13.36 | 0.0539 | 0.486 | 14.57 | 0.0327 | 0.1427 |
| EDM-Heun | 19.56 | 0.0699 | 0.422 | 12.66 | 0.0548 | 0.488 | 14.20 | 0.0332 | 0.1476 |
| RODS-CAS | 22.73 | 0.0661 | 0.400 | 16.08 | 0.0542 | 0.481 | 16.68 | 0.0318 | 0.1509 |

the sampling trajectory. To reduce computational cost while preserving performance, we employ a critical-step truncation strategy (Appendix B.5). We compare EDM-Euler, EDM-Heun, and our proposed method RODS-CAS. As shown in Figure 5, RODS-CAS achieves the lowest hallucination rate while correcting the highest proportion of faulty samples, and introduces no new hallucinations.

To provide a broader generation quality assessment, we have further computed standard image-quality and diversity metrics on 1,000 unconditional samples from FFHQ, AFHQv2, and 11k-hands. Table 1 reports Fréchet Inception Distance (FID) [18] (which quantifies the similarity between the generated and real image distributions in the Inception feature space), Inception Variance [37] (which measures variability in the latent representations), and DreamSim Diversity [17] (which captures the variance of features extracted from a batch of generated images). These results show that RODS-CAS preserves diversity while keeping FID comparable. Moreover, in our generation, the main subjects remain sharp and artifact-free.

These results, which spanning multiple datasets and varying task difficulty, demonstrate that our method generalizes effectively: it identifies potentially hallucinated samples at a controllable sensitivity; after applied robust corrections to the identified hallucination, it achieves over 25% correction for true positive depending on the difficulty of a given task, and preserves output quality even for false positives.

## 6   Conclusion

In this work, we proposed the equivalence between the diffusion sampling process and the numerical optimization continuation method. Build upon such optimization point-of-view, we introduced RODS, that is to detect and correct hallucinations in generative models. By leveraging geometric information from the loss landscape, our method adaptively adjusts the sampling direction in high-risk regions, significantly improving sample quality without additional training or excessive compute. We validated our approach across three datasets—AFHQv2, FFHQ, and 11k-hands. Overall, RODS-CAS delivers equally diverse, comparably high-quality unconditional samples while reducing hallucinations.

While RODS-CAS shows promising improvements over standard samplers in reducing hallucinations and correcting semantic inconsistencies, several aspects remain open for future study. First, our curvature-based detection provides reliable signals of instability, however, more precise localization could further improve correction accuracy. Empirically, we observe that early corrections tend to influence global semantics, while late ones mainly refine local details. This trade-off between early and late interventions offers flexibility, enabling RODS to adapt its correction strength to different scenarios. Second, our RODS algorithm can be directly implemented within the VE-ODE framework, which benefits from the equivalence between VE-based sampling and the continuation method under the usual parameterization. The extension of RODS to other sampling schemes (e.g., VP or latent-space diffusion) is conceptually straightforward but remains to be validated experimentally. Furthermore, its generalizability to text-to-image or conditional generation tasks should be straightforward but requires a thorough study in future work.

## Acknowledgment

This work was supported by the National Institutes of Health (NIH) under award number R01HL159183 and the National Science Foundation (NSF) AI Institute under award number 2112085.

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

# A   Theoretical Analysis

In this section, we provide theoretical justifications and analytical insights into the proposed method. Specifically, we draw a connection between the diffusion sampling dynamics and the continuation method, and further investigate the relationship between our proposed index and the local geometry of the score landscape, characterized by the Hessian's spectral properties.

## A.1   Continuation Method and sampling process matching

We formally derive the connection between diffusion sampling and continuation methods, showing that the sampling trajectory implicitly follows an energy descent path under gradually relaxed constraints. This view provides a principled interpretation of diffusion as a continuation process. While prior work has explored the link between Score-Matching Langevin Dynamics (SMLD) and optimization [9], our framework generalizes this connection to the diffusion sampling.

---

**Algorithm 3** Continuation Method

---

1: **Input:** Objective function $f(x)$, regularization term $R(x)$, initial $\lambda_0$, decay rate $\alpha$ ($\alpha < 1$), stopping criteria $\epsilon$
2: **Initialize:** $x^{(0)} \leftarrow$ initial guess, $k \leftarrow 0$
3: **while** $\lambda_k > \epsilon$ **do**
4:     Start from initial guess $x^{(k)}$, solve: $x^{(k+1)} \leftarrow \arg\min_x \left( f(x) + \lambda_k R(x) \right)$
5:     Update: $\lambda_{k+1} \leftarrow \alpha\lambda_k$, $k \leftarrow k + 1$
6: **end while**
7: **Output:** Final solution $x^{(k+1)}$

---

**Theorem 1 Proof**

*Proof.* **Case 1:** When $t = T$, or $T \to \infty$, the defined function can be simplified as:

$$
f_T(x) = -\log p_0(x) - \log \left\{ \sum_{i=1}^{K} \frac{w_i \mathcal{N}(x \mid \mu_i, \sigma_0^2)}{\sum_{j=1}^{K} w_j \mathcal{N}(x \mid \mu_j, \sigma_0^2)} \cdot \exp\left( \frac{\|x - \mu_i\|_2^2 \cdot T^2}{2\sigma_0^2(\sigma_0^2 + T^2)} \right) \right\}
$$

$$
= -\log \left\{ \sum_{i=1}^{K} w_i \mathcal{N}(x \mid \mu_i, \sigma_0^2) \cdot \sum_{i=1}^{K} \frac{w_i \mathcal{N}(x \mid \mu_i, \sigma_0^2)}{\sum_{j=1}^{K} w_j \mathcal{N}(x \mid \mu_j, \sigma_0^2)} \cdot \exp\left( \frac{\|x - \mu_i\|_2^2 \cdot T^2}{2\sigma_0^2(\sigma_0^2 + T^2)} \right) \right\}
$$

$$
= -\log \left\{ \sum_{i=1}^{K} w_i \mathcal{N}(x \mid \mu_i, \sigma_0^2) \cdot \exp\left( \frac{\|x - \mu_i\|_2^2 \cdot T^2}{2\sigma_0^2(\sigma_0^2 + T^2)} \right) \right\}
$$

$$
= -\log \left\{ \sum_{i=1}^{K} w_i \mathcal{N}(x \mid \mu_i, \sigma_0^2 + T^2) \right\}
$$

Thus:

$$
\min_x f_T(x) \equiv \min_x -\log p_T(x) \tag{9}
$$

where the resulting pure Gaussian distribution:

$$
p_T(x) = N(x \mid \mu, \sigma_0^2 + T^2), \quad \mu = \sum_{i=1}^{K} w_i \mu_i \tag{10}
$$

Thus, at $t = T$, minimizing $f_T(x)$ aligns with minimizing $-\log p_T(x)$, which confirms the behavior of the diffusion process in this case.

**Case 2:** When $0 < t < T$, the diffusion process probability distribution form can be expressed in this way:

$$
p_t(x) = \sum_{i=1}^{K} w_i \mathcal{N}(x \mid \mu_i, \sigma_0^2 + t^2), \tag{11}
$$

Then, its log form can be expressed as follow:

$$-\log p_t(x) = -\log \sum_{i=1}^{K} w_i \mathcal{N}(x \mid \mu_i, \sigma_0^2 + t^2)$$

$$= -\log \sum_{i=1}^{K} w_i \left\{ \frac{\sqrt{\sigma_0^2}}{\sqrt{\sigma_0^2 + t^2}} \cdot \mathcal{N}(x \mid \mu_i, \sigma_0^2) \cdot \exp\left( \frac{\|x - \mu_i\|_2^2 \cdot t^2}{2\sigma_0^2(\sigma_0^2 + t^2)} \right) \right\}$$

$$= -\frac{1}{2} \log \frac{\sigma_0^2}{\sigma_0^2 + t^2} - \log \left\{ \sum_{i=1}^{K} w_i \mathcal{N}(x \mid \mu_i, \sigma_0^2) \cdot \exp\left( \frac{\|x - \mu_i\|_2^2 \cdot t^2}{2\sigma_0^2(\sigma_0^2 + t^2)} \right) \right\}$$

$$= -\frac{1}{2} \log \frac{\sigma_0^2}{\sigma_0^2 + t^2} - \log \left\{ \sum_{i=1}^{K} w_i \mathcal{N}(x \mid \mu_i, \sigma_0^2) \cdot \sum_{i=1}^{K} \frac{w_i \mathcal{N}(x \mid \mu_i, \sigma_0^2)}{\sum_{j=1}^{K} w_j \mathcal{N}(x \mid \mu_j, \sigma_0^2)} \cdot \exp\left( \frac{\|x - \mu_i\|_2^2 \cdot t^2}{2\sigma_0^2(\sigma_0^2 + t^2)} \right) \right\}$$

$$= -\frac{1}{2} \log \frac{\sigma_0^2}{\sigma_0^2 + t^2} - \log \sum_{i=1}^{K} w_i \mathcal{N}(x \mid \mu_i, \sigma_0^2) - \log \left\{ \sum_{i=1}^{K} \frac{w_i \mathcal{N}(x \mid \mu_i, \sigma_0^2)}{\sum_{j=1}^{K} w_j \mathcal{N}(x \mid \mu_j, \sigma_0^2)} \cdot \exp\left( \frac{\|x - \mu_i\|_2^2 \cdot t^2}{2\sigma_0^2(\sigma_0^2 + t^2)} \right) \right\}$$

(The last two term is $\mathbb{E}_{p_0(x)} \exp\left( \frac{\|x - \mu_i\|_2^2 \cdot t^2}{2\sigma_0^2(\sigma_0^2 + t^2)} \right)$, describe the contribution proportion of each Gaussian distribution)

$$= -\frac{1}{2} \log \frac{\sigma_0^2}{\sigma_0^2 + t^2} - \log p_0(x) - \log \left\{ \sum_{i=1}^{K} \tilde{w}_i(x) \cdot \exp\left( \frac{\|x - \mu_i\|_2^2 \cdot t^2}{2\sigma_0^2(\sigma_0^2 + t^2)} \right) \right\}$$

$$(\tilde{w}_i(x) = \frac{w_i \mathcal{N}(x|\mu_i, \sigma_0^2)}{\sum_{j=1}^{K} w_j \mathcal{N}(x|\mu_j, \sigma_0^2)}, \sum_i \tilde{w}_i(x) = 1)$$

$$= -\frac{1}{2} \log \frac{\sigma_0^2}{\sigma_0^2 + t^2} + f_t(x)$$

Therefore, minimizing $f_t(x)$ over $x$ is equivalent to minimizing $-\log p_t(x)$ over $x$.

**Case 3:** When $t$ is 0, corresponding to the original distribution, the second term in the function vanishes. This simplifies our defined function to:

$$f_0(x) = -\log p_0(x) - \log \left\{ \sum_{i=1}^{K} \tilde{w}_i(x) \cdot \exp(0) \right\}$$

$$= -\log p_0(x) - \log \exp(0) - \log \left\{ \sum_{i=1}^{K} \tilde{w}_i(x) \right\}$$

$$= -\log p_0(x)$$

In this scenario, the function directly reflects the negative log-probability of the original distribution $p_0(x)$. This demonstrates that, in the absence of the regularization term, the function naturally converges to the original distribution, consistent with the diffusion process at its starting point.

Thus, the minimization of $f_t(x)$ for varying $t$ represents a gradual smoothing of the optimization landscape, analogous to the continuation method, where progressively subproblems are solved to reach the final target distribution. $\square$

**Theorem 2 Proof**

*Proof.* The proximal operator is defined as:

$$\text{prox}_{\eta f_t}(x_{t+1}) = \arg\min_y \left\{ f_t(y) + \frac{1}{2\eta} \|y - x_{t+1}\|_2^2 \right\}. \tag{12}$$

Taking the gradient of the objective function with respect to $y$ and setting it to zero gives:

$$\nabla f_t(y) + \frac{1}{\eta}(y - x_{t+1}) = 0, \tag{13}$$

which implies:
$$y = x_{t+1} - \eta \nabla f_t(y). \tag{14}$$

The diffusion sampling process updates $x_t$ using:
$$x_t = x_{t+1} - \eta \nabla f_t(x_{t+1}). \tag{15}$$

Comparing the two expressions, we see that:
$$x_t \approx \text{prox}_{\eta f_t}(x_{t+1}), \tag{16}$$

where the approximation arises from using $\nabla f_t(x_{t+1})$ in place of $\nabla f_t(y)$, which holds when $f_t(x)$ is sufficiently smooth. Thus, each diffusion sampling step corresponds approximately to solving the proximal operator. $\qquad\square$

## A.2 Understanding of the index

We analyze the theoretical underpinning of our proposed index by relating it to the maximum eigenvalue of the local Hessian of the score function. This analysis reveals that the index is inspired the local curvature, offering insights into the stability and reliability of the score field.

To intuitively understand when the learned score function becomes unstable, we look at how rapidly it changes in space. Let $v(x) := \nabla_x \log P_t(x)$ be the score function at time $t$, pointing in the direction of increasing data likelihood. In well-behaved regions, this score field changes smoothly. But in low-density areas, $v(x)$ can shift direction quickly—these are the regions where hallucinations are likely to occur.

Ideally, to capture how sharply $v(x)$ bends, we would examine its second derivative, a full Hessian tensor $\nabla_x \nabla_x v(x)$. However, this object is large and expensive to compute. Instead, we focus on the magnitude of the score function,
$$H(\|v(x)\|) = \nabla_x \nabla_x \|v(x)\|.$$

Rapid changes in this quantity often indicate instability in the local geometry of the score function. To measure how much this score magnitude could change under a small spatial shift $\delta$, we apply a first-order Taylor expansion:
$$\nabla_x \|v(x+\delta)\| - \nabla_x \|v(x)\| \approx H(\|v(x)\|) \cdot \delta.$$

This gives us a way to approximate how sensitive the model's confidence is to perturbations in input space. The Hessian norm $H$ can be used to compute the maximum change in the gradient due to the perturbation $\delta$. Especially for $\ell_2$ norm, it corresponds to the maximum eigenvalue $\lambda_1(H)$:
$$\|H\| = \max_{\|\delta\|=1} \frac{\|H\delta\|}{\|\delta\|} = \max_{\|\delta\|=1} \|H\delta\| \approx \max_{\|\delta\|=1} \|\nabla_x\|v(x+\delta)\| - \nabla_x\|v(x)\|\|$$

Finally, we measure the sensitivity of $v$ to changes in $x$ by computing the norm of the gradient difference:
$$\mathcal{H}(x) = \max_{\|\delta\|=1} \|\nabla_x\|v(x+\delta)\| - \nabla_x\|v(x)\|\|$$

However, directly solving this maximization over $\delta$ is computationally expensive. Instead, we approximate $\delta$ by the direction:
$$\delta = \arg \max_{\|\delta\|=1} \|v(x+\delta)\|.$$

By restricting the search to a local neighborhood of radius $\rho$, this yields the practical form of Eq. (8).

## B Ablation study and Visualization

This section provides additional analysis and visualizations to support the main results. We present ablation studies on key components of our method, including the choice of detection threshold and the impact of critical-step truncation. We further compare different strategies in solving $\delta$, report runtime statistics, and outline implementation details that improve efficiency. Finally, we describe the manual labeling protocol used for hallucination identification and provide additional visualizations.

## B.1 Dataset and Base Model

We evaluate our method on three benchmarks: AFHQ-v2 [26], FFHQ [26], and 11k-hands [1]. AFHQ-v2 contains high-resolution animal face images across three domains (cats, dogs, wild), with improved alignment and quality compared to the original AFHQ. FFHQ includes 70,000 high-quality human face images with rich variation in age, ethnicity, and lighting, making it a standard benchmark for face synthesis. The 11k-hands dataset contains 11,076 normal hand images ($1600\times1200$ pixels) from 190 individuals aged 18–75. Each subject was photographed opening both hands, from dorsal and palmar views, against a white background with consistent distance from the camera.

For the AFHQ and FFHQv2 datasets, we adopt the pre-trained models provided by EDM [25]. For the 11k-hands dataset, we train a diffusion model from scratch using the EDM framework, with input images resized to $128 \times 128$ and following standard training configurations.

Throughout our experiments, we utilize the Variance Exploding (VE) version of EDM with 40 sampling steps, which strikes a balance between generation quality and computational efficiency.

Training follows the default hyperparameters and architectural settings provided in the official EDM repository.

## B.2 Detection Threshold

In our current experiments, we observed that the choice of $\epsilon$ is influenced by both the dataset characteristics and the desired sensitivity level. For instance, 11k-hands features clean backgrounds and fine-grained hand details, so a smaller $\epsilon$ helps detect subtle hallucinations in localized regions. In contrast, face datasets (AFHQv2, FFHQ) often involve more complex backgrounds, where a slightly larger $\epsilon$ is preferred to avoid overreacting to natural variation in curvature.

We further analyze the impact of the detection threshold used in identifying hallucinations. As illustrated in Figure 6, the threshold controls a fundamental trade-off: a lower detection threshold increases sensitivity and allows the model to detect more hallucinated samples, but at the cost of falsely flagging some normal images as suspicious. In this experiment on the FFHQ dataset, we observe that a threshold of 0.09 achieves a true positive rate close to 80%, while keeping the false positive rate below 20%.

In practice, the threshold can be adapted based on the application scenario. For example, in high-risk domains such as medical imaging, a lower threshold may be preferable to ensure higher recall and minimize the chance of missing harmful hallucinations.

In the following ablation study, we address two key concerns related to false positives and efficiency: (1) Does mistakenly detecting a normal image as hallucinated degrade its quality after correction? Our experiments suggest that it does not. (2) Does correcting too many samples incur excessive computational overhead? We design a targeted correction strategy to mitigate this issue.

## B.3 Critical steps

As observed in Figures 13 and 14, the final differences between generated outputs often arise from a few *critical steps* during the sampling process. This suggests that whether or not a hallucination occurs is frequently determined by a small number of specific updates.

To verify and analyze this observation, we visualize, under fixed settings, the timesteps at which corrections are triggered across an entire batch. As shown in Figure 7, critical steps—i.e., steps where our proposed hallucination index exceeds the threshold for at least one image in the batch—tend to concentrate in the middle stage of the trajectory. This pattern is consistent across tasks and datasets, and aligns well with our hypothesis that mid-range steps are most influential in determining sample quality.

## B.4 Different Strategies in Solving $\delta$

Solving the inner maximization exactly can be costly, especially in high dimensions. A simple and efficient alternative is to approximate the direction using a first-order step. If $\mathcal{M}$ is an $\ell_2$-ball of radius $\rho$, then:

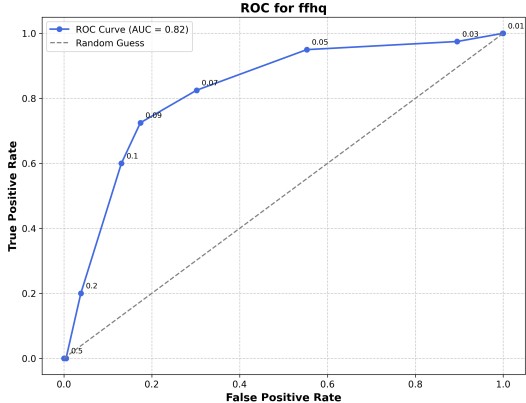

Figure 6: ROC curve for hallucination detection on the FFHQ dataset. Each point corresponds to a different detection threshold. Lower thresholds yield higher true positive rates but also increase false positives. For example, a threshold of 0.09 achieves ∼80% TPR at under 20% FPR. Thresholds can be adjusted depending on the application's sensitivity to hallucination risk.

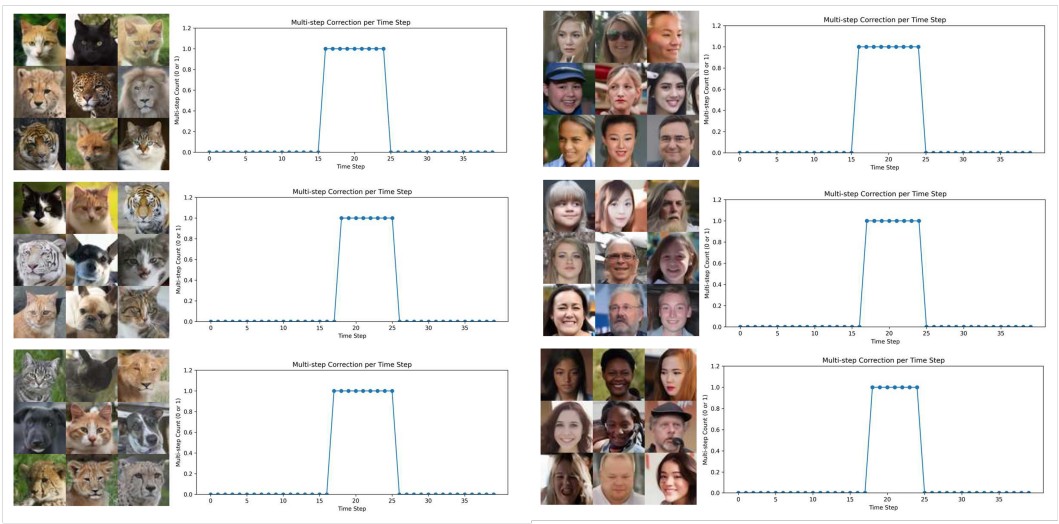

Figure 7: Detection of critical steps. Each Left: a batch of generated samples. Each Right: for each timestep, we mark it as "critical" (value 1) if the proposed detection index exceeds the threshold for at least one image in the batch. This highlights which steps are most likely to trigger robust corrections.

- **SAS:**

$$\delta_{i,\text{SAS}}^* \approx \rho \cdot \frac{\nabla f_{t_i}(x_i)}{\|\nabla f_{t_i}(x_i)\|}.$$

Since $\nabla f_{t_i}(x_i) \approx -\nabla \log p_{t_i}(x_i)$, you may flip the sign depending on the implementation.

- **CAS:**

$$\delta_{i,\text{CAS}}^* \approx \rho \cdot \frac{\nabla \|\nabla f_{t_i}(x_i)\|}{\|\nabla \|\nabla f_{t_i}(x_i)\|\|}.$$

Again, it's common to substitute $\nabla f_{t_i}(x_i) \approx -\nabla \log p_{t_i}(x_i)$ here as well.

Another natural idea is to solve $\delta$ using multiple iterations of gradient descent. To investigate this, we compare our default single-step update with 5-step and 10-step gradient descent variants. As illustrated in Figure 8, we observe that for the vast majority of cases, the results across all three strategies are nearly identical—suggesting that additional optimization steps do not significantly alter the output. Specifically, only a small fraction of images exhibit perceptible differences between

Table 2: Hallucination analysis on the 11k-hands dataset using various samplers. Our method achieves the highest correction rate without introducing new hallucinations, while also maintaining low inference time.

| Method | Hall.%↓ | Correction%↑ | New Hall.%↓ | Time (s) |
|---|---|---|---|---|
| EDM-Euler | 19.4% | N/A | N/A | 2.71 |
| EDM-Heun | 18.0% | 17.7% | 2.5% | 5.38 |
| Euler-200 | 16.9% | 18.9% | 1.4% | 13.76 |
| RODS-CAS | 14.3% | 26.3% | 0.0% | 4.82 |

single-step and multi-step updates (3 out of 184 on AFHQ, and 4 out of 222 on FFHQ), as shown on the right-hand side of the figure.

Given the negligible improvement and substantially increased computational cost, we recommend using the single-step update in practical applications.

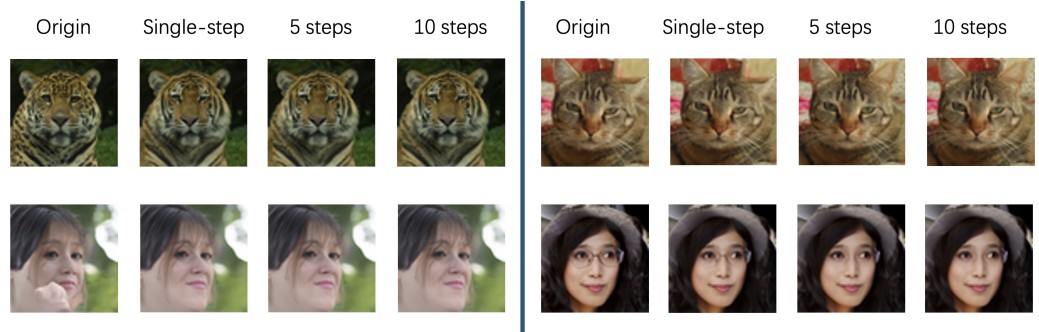

Figure 8: Comparison between single-step and multi-step gradient updates (5 steps, 10 steps). Left: in most cases, all update strategies lead to visually identical results. Right: rare examples where multi-step updates differ from the single-step version (3/184 in AFHQ, 4/222 in FFHQ). Given the marginal benefit and higher computational cost, we recommend the single-step strategy.

## B.5   Acceleration Strategy and Running Statistics

Building on the previous analysis, we observe that most hallucination-inducing steps occur in the middle of the sampling trajectory, and that single-step correction is sufficiently effective. For datasets with a high hallucination rate, we apply single-step correction to all samples. To improve inference efficiency, we restrict the application of correction sampling to the middle portion of the trajectory—specifically, from 10% to 50% of the total sampling steps. This selective strategy preserves the effectiveness of hallucination correction while reducing the risk of quality degradation from excessive perturbations.

We compare EDM-Euler, EDM-Heun, and our proposed method RODS-CAS. For a fair comparison, we also include EDM-Euler with 200 sampling steps. As shown in Figure 9, 10 and Table 2, our method achieves the highest hallucination correction rate without introducing new artifacts, while maintaining a low inference time.

This result also explains why simply reducing the step size is insufficient. Even with 200 uniform steps, which is five times smaller than the 40-step baseline, the hallucinations persist. In contrast, RODS-CAS adaptively adjusts its update direction in high-curvature regions and eventually removes artifacts. The key issue lies in directional error within low-density, high-uncertainty areas; smaller steps cannot fix a wrong direction, while curvature information helps steer sampling toward stable regions.

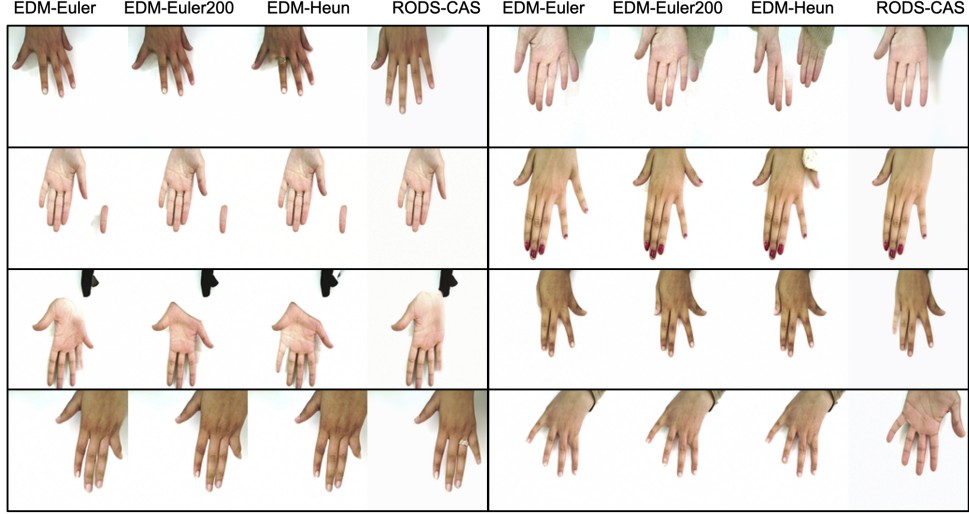

Figure 9: Examples where our method successfully reduces hallucinations that are not corrected by stronger solvers such as EDM-Euler (200 steps) and EDM-Heun (40 steps).

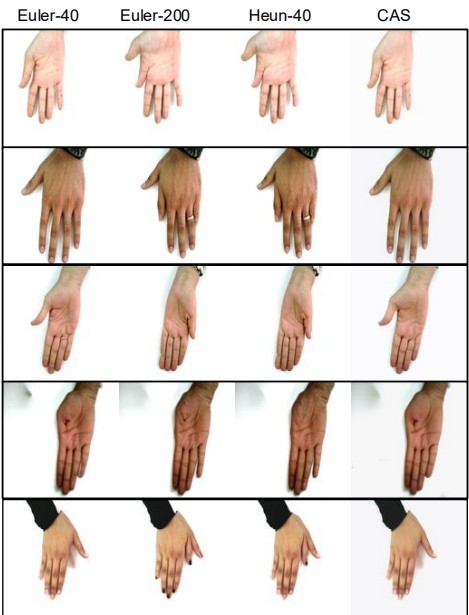

Figure 10: Examples where advanced solvers (Euler-200, Heun-40) introduce new hallucinations compared to Euler-40, while our method remains stable.

## B.6 Labeling Detail

Since there is no universally agreed-upon definition of hallucination in generative models, and no established gold standard for evaluation, we resort to manual annotation to assess the presence of hallucinations. Given that the identification of visual hallucinations in our setting does not require domain-specific expertise, we recruited lab members to perform the labeling.

For face hallucinations on AFHQv2 and FFHQ, hallucinations are identified and annotated based on human perception, common sense, and visual plausibility. For hand hallucinations on the 11k-hands

dataset, each generated sample is manually assigned to one of the following seven categories: (1) normal, (2) extra fingers, (3) missing fingers, (4) incorrect finger structure, (5) abnormal palm, (6) multiple hands, and (7) unrecognizable or implausible. This taxonomy allows for a fine-grained analysis of hallucination types and facilitates more targeted evaluation and correction.

To further improve evaluation precision, we conducted pairwise human comparisons on anonymized image pairs (RODS vs. baseline). Unlike binary hallucination labels, these pairwise judgments capture relative image quality, providing a more nuanced measure of generation fidelity. The evaluation rubric is as follows:

- **Better**: The RODS-generated image is perceived as more faithful or realistic, often due to improved correction of visual anomalies (e.g., fixing a misplaced eye) without introducing new defects.

- **Worse**: The RODS-generated image is clearly degraded relative to the baseline, typically because it introduces new artifacts or hallucinated elements that were not present in the original.

- **Same**: No human-identifiable difference exists between the baseline and the RODS-generated image.

- **Unclear**: A change occurs, but it does not involve new hallucinations, and it is difficult to judge which version is better (e.g., slight pose changes or missing accessories like a ring).

All images were labeled independently by two human raters to ensure objectivity. Whenever their judgments diverged, the pair reviewed the cases together and discussed until they reached a consensus, producing a single, agreed-upon label for each image. This consensus process both minimized individual bias and improved the overall reliability of our ground-truth data.To evaluate RODS-CAS's impact relative to the baseline (EDM-Euler), annotators first identified hallucinations in the Euler-sampled outputs. They then compared each corrected image against its Euler counterpart, categorizing the result as "better," "worse," "same," or "unclear". This side-by-side comparison made it easy to see whether our method truly reduced artifacts, or introduced new ones. Examples of annotated hallucinations can be found in Figure 11 (faces) and Figure 12 (hands).

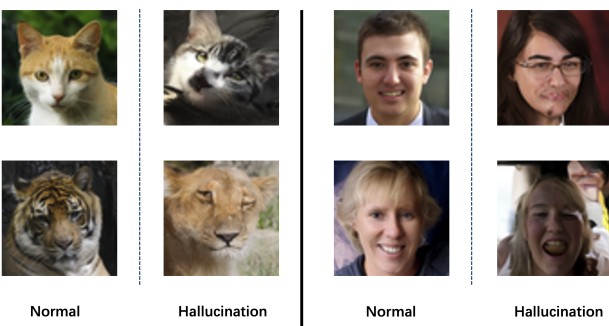

Figure 11: Examples of face hallucinations on AFHQv2 (left) and FFHQ (right). Hallucinations are identified and labeled based on human perception, common sense, and visual preferences.

## B.7  More Visualization

Figures 13 and 14 show the generation trajectories for the same batch of samples from the AFHQ and FFHQ datasets under different sampling methods. Several interesting observations can be made. For most well-formed samples, our method produces trajectories and outputs that are visually indistinguishable from the baseline. In contrast, for samples that eventually exhibit hallucinations, deviations in the trajectory often emerge during the middle stages of sampling, suggesting that a few critical steps play a decisive role. A more rigorous analysis of this phenomenon is provided in earlier sections.

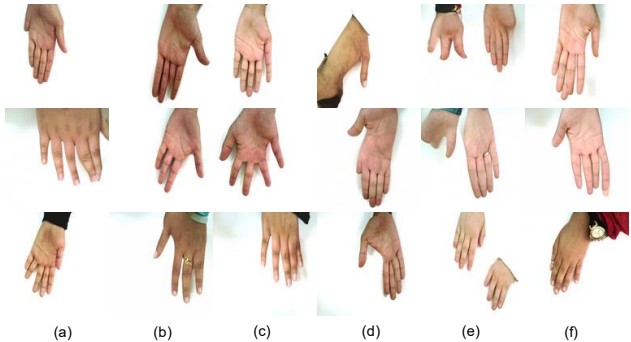

Figure 12: Examples of hand hallucination types produced by Euler-40 sampling. From left to right: (a) Extra fingers, (b) Missing fingers, (c) Incorrect finger structure (correct number but anatomically implausible), (d) Abnormal palms, (e) Multiple hands, and (f) Other distortions or unrecognizable features.

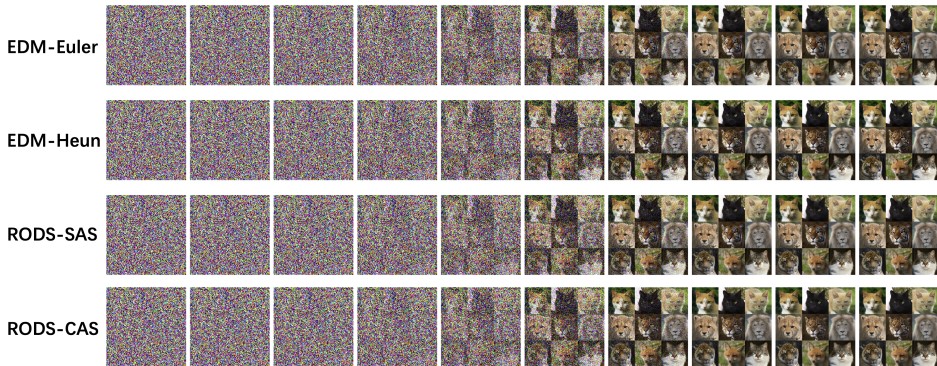

Figure 13: Generation trajectories on the AFHQv2 dataset from a fixed random seed and batch (batch size = 9). Each row corresponds to a different method, and columns depict intermediate samples $x_t$ from timestep $t = T$ (right) to $t = 0$ (left). Our method selectively modifies poor-quality generations—such as correcting the distorted eyes of the cat in the top-right corner—while preserving images without hallucinations unchanged.

## C    Literature Review on Hallucination

### C.1    Definition of Hallucination

**Hallucination** in generative AI refers to the phenomenon where a model produces content that is **unfounded, unfaithful**, or **not grounded** in the provided input or in reality [30, 24]. In natural language generation, this often means the model outputs nonsensical or factually incorrect statements that were never in the source material [41, 14]. The term has been extended to visual and multimodal AI: for example, in image captioning or vision-language tasks, it denotes instances where the model "hallucinates" objects or details that are not actually present in the input image [42].

Hallucination is broadly recognized as a critical problem for **trustworthy and safe AI**. It undermines users' trust in AI systems and poses serious risks in real-world applications. Models that hallucinate make content unreliable and unpredictable, which is especially problematic in **high-stakes domains** [24, 28]. In the healthcare context, the consequences can be life-threatening: for instance, a medical question-answering system or summarization model that fabricates a nonexistent symptom or an incorrect medication instruction could mislead clinicians or patients [24].

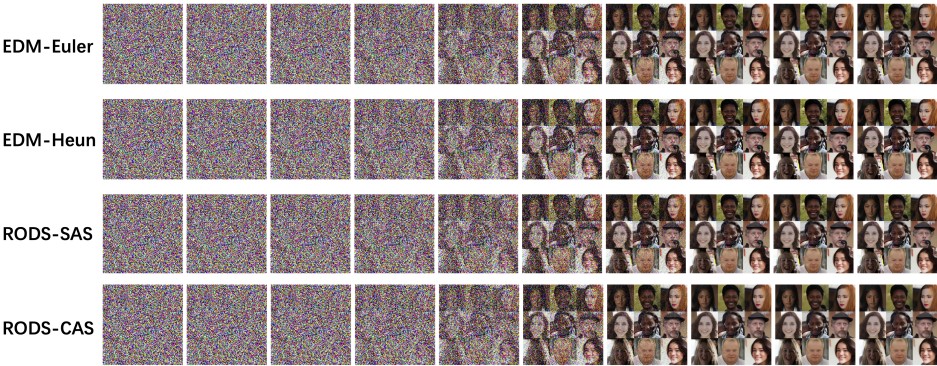

Figure 14: Generation trajectories on the FFHQ dataset from a fixed random seed and batch (batch size = 9). Each row corresponds to a different method, and columns depict intermediate samples $x_t$ from timestep $t = T$ (right) to $t = 0$ (left). Our method selectively refines images containing hallucinations—such as unnatural artifacts near the mouth, or unrealistic eyebrows and hair—while leaving images without hallucinations unchanged.

## C.2 Hallucination in Image Generation

In the context of image generation, a hallucination typically means the model has created **visual artifacts** or **implausible objects** in generated images. For example, diffusion models like DALL·E or Stable Diffusion sometimes produce people with anatomically impossible features (such as extra fingers or distorted limbs) or blend background elements in incoherent ways [2, 38, 8]. From a technical standpoint, such hallucinations correspond to the model sampling from **outside the true distribution** of the training data. In other words, the generator is creating images that look superficially plausible but contain details that no real image would have, hence indicating the model has generalized in an incorrect, unrestricted manner.

In **scientific** and **medical** domains, the stakes are extremely high because a hallucinated visual detail can lead to false scientific conclusions or diagnostic errors. In medical imaging, diffusion models trained primarily on healthy data may hallucinate away tumors or introduce fictitious lesions, risking dangerous misdiagnoses, as highlighted by [27] denoted structural hallucination. In scientific domains such as satellite imagery, models may fabricate false visual features—e.g., showing floodwater in dry regions—potentially misleading researchers and policymakers [11]. In both cases, hallucinated details not only degrade visual fidelity but also threaten the integrity and safety of downstream decisions, underscoring the urgent need to mitigate hallucinations in high-stakes applications [28].

## C.3 Reduce Hallucination in Diffusion Model

Although hallucination is not a well-defined phenomenon—making its detection and suppression inherently difficult—it remains a central concern for deploying generative AI in high-stakes domains. Recent research has thus focused on developing techniques to address hallucinations during the generation process, particularly in diffusion models, which now dominate image synthesis tasks. Several promising directions have emerged:

**Trajectory-based consistency control.** Aithal et al. [2] observe that hallucinated samples often exhibit high prediction variance in the reverse diffusion trajectory, signaling off-manifold behavior. They propose a simple but effective variance-based metric to filter out such aberrant samples. Huang et al. [22] regularizes the sampling trajectory by constraining the norm of the predicted noise to remain aligned with the score direction, ensuring update validity within high-confidence regions.

**Local diffusion with OOD region isolation.** Kim et al. [27] tackle structural hallucinations in conditional diffusion by partitioning inputs into in-distribution (IND) and out-of-distribution (OOD) regions. An anomaly detector produces a probabilistic OOD mask, and two separate diffusion branches process IND and OOD content, respectively, followed by a fusion step.

**Early stopping of diffusion decoding.** Tivnan et al. [51] propose truncating the sampling process before full convergence to avoid overfitting to noise and hallucinating unnecessary details. Their

results show that early stopping improves structural fidelity without sacrificing perceptual quality. They further introduce a "Hallucination Index" to quantitatively evaluate model faithfulness.

**Attention Modulation Adjustment.** Oorloff et al. [39] propose Adaptive Attention Modulation (AAM) to reduce hallucinations in diffusion models by dynamically adjusting self-attention behavior during inference. They apply masked perturbations to disrupt early-stage noise that may propagate into hallucinations. AAM directly modulates the internal attention layers, providing a lightweight way to suppress hallucinated artifacts while preserving image quality.

**Structure-specific fine-tuning.** Several works refine diffusion models in specific hallucination-prone regions. For hand generation, HandRefiner [32] propose to use fine-grained annotations and localized losses to correct anatomical distortions. DreamBooth-style fine-tuning [44] enable localized hallucination correction with minimal retraining.

In summary, these complementary strategies contribute to a growing toolkit for hallucination mitigation in diffusion models. Trajectory-based filtering enables lightweight real-time rejection, local diffusion separates difficult regions for targeted reconstruction, early stopping avoids overgeneration, and structure-specific finetuning enables precise corrections in vulnerable subregions. However, hallucination detection and mitigation remain an open problem: different methods target different manifestations of hallucination, and the lack of a unified definition or standardized benchmark continues to pose challenges for systematic evaluation and comparison.

# D   Literature Review on Bridging Optimization and Diffusion Models

This section reviews emerging perspectives that connect diffusion models with optimization. Prior work relates score-based sampling to stochastic gradient methods, uses diffusion as an optimizer in planning and inverse problems, and explores alternative views like Schrödinger bridges. Our approach builds on these by framing diffusion as a continuation method, offering new insights and opportunities for algorithmic improvement.

## D.1   Score-Matching Langevin Dynamics as Stochastic Gradient Descent

Early work on Score-Based Generative Models explicitly drew parallels to simulated annealing, introducing annealed Langevin dynamics that uses large noise (high "temperature") initially and decreases it step by step [49]. At a fixed noise level, **Score-Matching Langevin Dynamics (SMLD)**, each step is essentially a **stochastic gradient ascent (SGD)** on the log-density of an intermediate distribution (using the learned score $\nabla_x \log p_t(x)$) with added Gaussian noise for exploration. This is precisely Langevin Monte Carlo, which mirrors gradient descent but injected with noise to sample rather than converge to a single mode.

In fact, Langevin sampling can be seen as a close cousin of stochastic gradient descent in optimization: the only difference is the noise term that prevents collapse to a mode. Recent research has made this connection explicit. [52] point out the "high connection" between diffusion's sampling process and SGD, and import SGD techniques into diffusion. They introduce an adaptive momentum sampler, analogous to adding a heavy-ball momentum term to Langevin dynamics.

Similarly, other works have looked at second-order optimization ideas in diffusion: e.g. preconditioning the Langevin updates using curvature information. [50] derives an optimal preconditioner for Langevin MCMC based on the Fisher information matrix, effectively a Newton-like method that adapts step sizes to the geometry of the target distribution. Such curvature-aware updates amount to using an approximate Hessian to accelerate sampling, paralleling Newton's method in optimization.

## D.2   Solving Optimization as Diffusion Sampling

Diffusion models are deeply connected to energy-based generative modeling (EBM), and this link further cements the optimization interpretation. An EBM defines an energy function $E(x)$ (or unnormalized log-density); sampling from it typically involves running MCMC (e.g. Langevin dynamics) – essentially gradient descent on $E(x)$ with noise. Score-based diffusion models, instead of specifying a single energy, learn a time-dependent score function $s_\theta(x, t) \approx \nabla_x \log p_t(x)$ for each noise level $t$ [47].

Beyond improving diffusion itself with optimization techniques, a complementary line of research uses diffusion models as optimization solvers for the energy-based objective function or noised objective function. [23] reframes offline reinforcement learning as a diffusion-based trajectory generation problem. Diffuser is a denoising diffusion probabilistic model that plans by iteratively refining noise into a coherent trajectory.

In essence, the diffusion model acts as an optimizer in the space of trajectories, finding plans that both resemble the training data distribution and achieve the desired goal (via guidance). Follow-up works have expanded on this concept. For example, [21] decomposes the diffusion planning into two stages: (1) quickly generating a feasible rough trajectory (using an autoregressive policy model), and (2) diffusion-based trajectory optimization to refine it for higher quality. [53] introduce safety or constraints into diffusion planners. [35] further advances the idea by leveraging value functions as energy surrogates, enabling efficient trajectory optimization without relying on sampling from the optimal policy.

Overall, in the decision-making domain, diffusion models a powerful framework to **solve optimization problems** over sequences of actions by sampling from a progressively refined distribution. The stochastic, continuation-style generation (from noise to solution) helps avoid local optima in planning, much like **diffusion sampling process**.

### D.3 Diffusion sampling as continuation method

Our work aligns more closely with the first category but distinctly frames diffusion sampling as a continuation or homotopy method from numerical optimization. Unlike conventional interpretations based on fixed-noise-level SMLD, the continuation (homotopy) view of diffusion sampling treats the generation as solving a sequence of gradually changing optimization problems. At $t = T$ (maximum noise), the "objective" is trivial (the target distribution is pure Gaussian noise, which the sampler can produce easily). As $t$ decreases, the implicit objective continuously deforms, introducing more data structure; by $t = 0$, it becomes the true data distribution.

Viewing diffusion sampling through the lens of continuation methods not only provides conceptual clarity, but also opens up promising directions for future algorithmic improvements. By treating each diffusion step as a progressively refined subproblem, this perspective aligns with a wide range of techniques in numerical optimization. For example, it motivates adaptive step size scheduling, trust-region-based stability control, and convergence and error analysis. Our proposed robust optimization formulation is inspired by this perspective, while other aspects are left for future exploration.

### D.4 Other related point of view

Beyond the continuation and optimization viewpoints discussed above, several other perspectives have also been proposed to connect diffusion models with optimization.

One such perspective is the Schrödinger bridge formulation, which seeks the most likely stochastic process (via a controlled drift) that transforms a prior distribution into the data distribution over time. This leads to an entropy-regularized optimal transport problem, solvable via iterative projections—a form of optimization in probability space. Several works reinterpret score-based diffusion as a Schrödinger bridge problem, thereby establishing a formal connection between diffusion models and stochastic control theory [15].

Another active research direction explores the use of diffusion models for solving inverse problems in imaging and related domains. Diffusion models serve as powerful learned priors, offering a structured way to sample from complex data distributions. Recent methods adopt a plug-and-play approach, where diffusion-based generation is alternated with steps that enforce data fidelity. For example, Chung et al. [12] propose Diffusion Posterior Sampling (DPS) for general noisy inverse problems, which combines denoising steps with gradient-based corrections based on measurement likelihood. At each diffusion timestep, the sample is not only denoised but also guided toward satisfying the observed data—e.g., using gradients of the log-likelihood $\nabla_x \log p(y|x)$—resembling a projection onto the data-consistent manifold.

