# OpenReview forum: "RODS: Robust Optimization Inspired Diffusion Sampling for Detecting and Reducing Hallucination in Generative Models"
_NeurIPS.cc/2025/Conference — NeurIPS 2025 poster_

### Official Review · Reviewer_UPcW · 2025-07-01

**Clarity:** 3
**Significance:** 2
**Originality:** 2
**Rating:** 5
**Confidence:** 4

**Summary:**

The authors introduce an inference-time method (RODS) to reduce the number of hallucinations in diffusion models. In particular, if the curvature index $\mathcal H(x)$ of a particular sample at a particular diffusion time exceeds some hyperparameterized value $\epsilon$, RODS uses a guidance-like term $\delta = \arg\max_{||\delta|| = \rho} ||v(x+\delta)||$ to guide the sample towards stability. The method is validated over AFHQv2, FFHQ, and 11k-hands and shows good performance.

**Questions:**

See weaknesses. Would love to see further comparison with other methods.

Additionally, qualitatively examining Figure 4, RODS-CAS clearly achieves fewer hallucinations, but it also results in the background becoming blurrier. Is there an explanation for this phenomenon?

**Ethical Concerns:**

["NO or VERY MINOR ethics concerns only"]

**Final Justification:**

My main notes were regarding other works, to which they have provided satisfactory explanations / plans to address, as well as the  Overall the method is novel and achieves fewer hallucinations across multiple datasets. Therefore, I am increasing my rating to accept.

**Limitations:**

No real discussion of limits of current method on current task, only discussion of how it was not yet applied to another task (conditional generation).

**Paper Formatting Concerns:**

None spotted.

**Quality:**

3

**Strengths And Weaknesses:**

Strengths:
1. Paper is clear and well written.
2. Good performance on reducing hallucinations in AFHQv2, FFHQ, and 11k-hands.
3. Method is relatively efficient in inference time.

Weaknesses:
1. Similar idea using norms to dynamically adjust diffusion sample present in other paper [1].
2. Not enough evaluations of other hallucination reducing methods, e.g. [2], [3]

[1] Constrained Diffusion with Trust Sampling.
[2] Mitigating Hallucinations in Diffusion Models through Adaptive Attention Modulation.
[3] Understanding Hallucinations in Diffusion Models through Mode Interpolation.

---

> ### Author Rebuttal · Authors · 2025-07-31
>
> ### **W&Q1: Related works and Comparison**
> Thank you for pointing out these related works. We will include citations and a brief discussion of [1]–[3] in the revised version.
>
> While [1] ("Constrained Diffusion with Trust Sampling") uses the norm of the predicted noise, aligned with the score direction, as a constraint for update validity, our approach instead focuses on the norm of local changes in the score field to guide updates. Given the differing objectives and mechanisms, a direct comparison is not applicable.
>
> Regarding [2] (“Mitigating Hallucinations through Adaptive Attention Modulation”), it adopts a different strategy—modulating attention during sampling—while our approach focuses on trajectory-level geometric regularization. As the paper was posted on arXiv in February 2025 and no code was available at the time of our submission, a direct experimental comparison was not feasible.
>
> As for [3] (“Understanding Hallucinations through Mode Interpolation”), we do include a comparison in Section 4.1 and Appendix C.3. Their method detects hallucinations via **temporal instability** along the denoising trajectory, whereas our approach targets **spatial-domain instability** and further enables **correction**. Additionally, their released code (for now) only covered 1D&2D Gaussian and SIMPLE SHAPES dataset, which could not be naturally transferred to our datasets. We plan to provide a fair experimental comparison once their full implementation becomes available.
>
> We appreciate the reviewer’s emphasis on thorough comparison. As noted in Section 5, due to substantial differences across prior works in methodology, assumptions, and evaluation protocols, there is currently no standardized benchmark for hallucination detection and correction. We are committed to releasing our code and annotated data to facilitate future research and establish a common basis for meaningful comparison across methods.
>
> ---
>
> ### **Q2: Evaluation on background blurry problem**
>
> Thank you for pointing out the background blurrier phenomenon. This occurs because it is hard to distinguish between **high-frequency textures and hallucinations**. In our generation, the main subjects remain sharp and artifact-free, which is precisely the goal of our correction. The slight background smoothing does not compromise the diversity of the generation or the sharpness of the primary objects.
>
> To validate such idea, we have further computed standard image-quality and diversity metrics on 1,000 unconditional samples from FFHQ, AFHQv2, and 11k-hands. The table below reports FID, Inception Variance [1] (which quantifies variability in the latent representations), and DreamSim Diversity [2] (which measures the variance of features extracted from a batch of generated images, and our batch size is 64).
>
>
> | Dataset  | FFHQ                |        |         | AFHQ2               |        |         | 11k-hands        |        |         |
> |----------|---------------------|--------|---------|----------------------|--------|---------|-------------------------|--------|---------|
> |          | FID ↓               | Inception-Var ↑ | DreamSim ↑ | FID ↓               | Inception-Var ↑ | DreamSim ↑ | FID ↓               | Inception-Var ↑ | DreamSim ↑ |
> | EDM-Euler    | 20.48              | 0.0669 | 0.407   | 13.36              | 0.0539 | 0.486   | 14.57          | 0.0327                    | 0.1427|
> | EDM-Heun      | 19.56          | 0.0699 | 0.422 | 12.66          | 0.0548 | 0.488 | 14.20          | 0.0332                    | 0.1476               |
> | RODS-CAS      | 22.73              | 0.0661 | 0.400   | 16.08              | 0.0542 | 0.481   | 16.68          | 0.0318                    | 0.1509               |
>
> These results show that RODS-CAS preserves diversity while keeping FID comparable. In our generation, the main subjects remain sharp and artifact-free. **_This is further supported by the 11k-hands experiment, where our method improves hand confidence (HAND-CONF) from 0.9216 (EDM-Euler) and 0.9227 (EDM-Heun) to 0.9347 (RODS-CAS)._**  HAND-CONF is computed by running a pre-trained hand detector [3] and calculating the confidence scores of the detections. This metric is widely used to assess the anatomical plausibility of generated hands. Because hands are structurally well-defined and challenging to synthesize, this benchmark provides a sensitive and interpretable proxy for visual realism.
>
>
> [1] Miao, Zichen, et al. "Training diffusion models towards diverse image generation with reinforcement learning." CVPR. 2024.
> [2] Domingo-Enrich, Carles, et al. "Adjoint matching: Fine-tuning flow and diffusion generative models with memoryless stochastic optimal control." arXiv preprint 2024.
> [3] Zhang, Fan, et al. "Mediapipe hands: On-device real-time hand tracking." arXiv preprint 2020.
>
> ---
>
> ### **L: Limitation on current dataset**
>
> Thank you for raising this up. In the unconditional generation scenario, we have tested RODS on well-established benchmarks (AFHQv2, FFHQ, 11k-hands), where hallucinations can be easily measured without domain expertise. We acknowledge that in highly diverse or specialized domains, such as medical imaging, the definition of “hallucination” must be grounded in expert judgment. We are collaborating with domain practitioners to evaluate RODS under these conditions.

---

> ### Author Response · Authors · 2025-08-06
>
> Dear Reviewer UPcW,
>
> Thank you for your thoughtful review and for the time you’ve invested in our paper. We’ve gone through your feedback and responded to each point in the rebuttal above.
>
> Please let us know whether our responses have resolved your concerns, especially with the discussion period ending in two days. If anything remains unclear or you’d like further clarification, we’d be happy to continue the discussion. And if our responses have resolved your concerns, we would greatly appreciate your reconsideration of our score.

---

> > ### Comment · Reviewer_UPcW · 2025-08-07
> >
> > Thanks for your response! Appreciate the notes on related works and the additional experiment. No further questions.

---

### Official Review · Reviewer_ZPMe · 2025-07-01

**Clarity:** 2
**Significance:** 2
**Originality:** 3
**Rating:** 4
**Confidence:** 2

**Summary:**

The paper views the reverse diffusion process as a continuation scheme, where each timestep solves an increasingly ill-conditioned optimization subproblem. Building on this perspective, the authors integrate a curvature-based detector into standard samplers to identify regions where the score may be unreliable, and apply a worst-case-aware correction to keep the trajectory stable, without requiring retraining. This approach reduces hallucinations, as validated by experimental results.

**Questions:**

1. Is there any theoretical justification for why sharp curvature changes indicate high-risk regions during sampling?
2. Is there any analysis supporting the idea that minimizing the worst-case outcome (as in RODS) can give overall more robust sampling? Under what scenarios is this true?
3. Just to confirm my understanding: is the core idea behind Algorithm 2 to minimize the objective shown after line 181, when $\mathcal{F}_t$ is in the form of the expression after line 219?
4. The experiment results focus on hallucination. Is there any evaluation of overall generation quality, e.g., FID or similar metrics, to ensure RODS doesn’t degrade sample quality?

**Ethical Concerns:**

["NO or VERY MINOR ethics concerns only"]

**Final Justification:**

I thank the authors for their detailed response that addressed my questions and concerns. I would like to increase my recommendation score to 4.

**Limitations:**

yes

**Quality:**

2

**Strengths And Weaknesses:**

### Strengths
1. The paper introduces a novel mathematical perspective on diffusion sampling by framing it as a continuation method.
2. The work is fairly complete: combining theoretical insight with a practical algorithm that is both lightweight and easy to integrate. The proposed method demonstrates meaningful empirical improvements across several datasets.

### Weaknesses
To me, the main value of the paper is in the theoretical viewpoint, while the experiments mostly serve to support and illustrate that idea. I do wish the paper leaned a bit more into the theory: right now, the mathematical part is quite straightforward and mainly used to justify the optimization interpretation. But it doesn’t really explain why or when RODS should reduce hallucinations. It would be more convincing if there were some analysis showing, for example, that under certain conditions on the score function or the data distribution, RODS would provably outperform standard samplers. That kind of result would help bridge the gap between the conceptual contribution and the algorithmic claims.

---

> ### Author Rebuttal · Authors · 2025-07-31
>
> ### **W: Explain why / when RODS reduce hallucinations**
> Thank you for pointing this out. Below we summarize the key take‑aways and point to the detailed explanations we have now added (see the two responses to the questions below).
>
> 1) Sharp local curvature ⇒ high score uncertainty ⇒ higher hallucination risk (**Q1**)
>
> 2) Our look‑ahead worst‑case update provably lowers the maximum one‑step error in exactly those high‑risk zones while (i) score locally Lipschitz, (ii) score error bounded by $\rho$, (iii) step stays inside the ball. (**Q2**)
>
> ---
>
> ### **Q1: Connection between Curvature, Uncertainty, and Hallucination**
>
> Thank you for asking us to clarify the connection between curvature, uncertainty, and hallucination. The key idea is that **hallucination correlates with model uncertainty** [1], which can be quantitatively captured by the **“rapid fluctuations (curvature)”** of the density field.
>
>
> **Curvature is a proxy for model uncertainty**
>
> - **Score field.**  We denote the learned score by $v(x) = \nabla_x \log P_t(x)$.
> - **Curvature index.**  As stated in §4.1 (Eq. 8), we define our curvature index is:
>
> $$
> H(x) = \|\nabla_x \|v(x + \delta)\| - \nabla_x \|v(x)\|\|,
> \quad
> \delta = \arg \max_{\|\delta\| = \rho} \|v(x + \delta)\|,
> $$
>
> Because $H(x)$ compares the gradient of $\|v\|$ at two nearby points, it measures **second-order variation** of the density field, i.e., how smoothly the score field changes in space. This reflects the key idea that **hallucination correlates with model uncertainty** [1].  By probing these rapid, spatial fluctuations, i.e., second‑order changes or “curvature”, we obtain a quantitative signal for potential sampling failures. In Appendix A.2, we further describe the conceptual foundation in detail:
>
> > “In well-behaved regions, this vector field changes smoothly. But in low-density areas, $v(x)$ can shift direction quickly—these are the regions where hallucinations are likely to occur.” (line 467–471)
>
> > “Rapid changes in this quantity often indicate instability in the local geometry of the score function.” (line 475–476)
>
>
> ---
>
> ### **Q2: Connection between optimization against the worst-case scenario and robustness**
>
> This follows a common principle in robust optimization: by guarding against the **worst-case scenario**, we ensure **robust and reliable performance** across all admissible uncertainties. If loss can be reduced even under the most adverse local perturbation, then all other nearby cases are guaranteed to be no worse. This approach is valid when the score function is locally smooth, the uncertainty is bounded, and the step size remains small. Below, we provide a detailed analysis to explain why it enhances robustness and under what conditions the guarantee holds.
>
> 1. **What the algorithm actually does**
>
> | Step | What we do | Why we do it |
> |------|------------|--------------|
> | **Search** | Look around the current sample $x$ inside a small radius and pick the neighbour $\hat{x}$ where the score network is least reliable. ‑ **SAS** : choose the point with the largest score magnitude (most likely to overshoot). ‑ **CAS** : choose the point whose score changes fastest (quickest direction flip). | Identify the spot most prone to error. |
> | **Descent at $\hat{x}$** | Compute the steepest descent direction $d$ *at that worst point*. | Neutralise the largest local error. |
> | **Update the $x$** | Move the original sample a small step along the safe direction just computed. | Apply the error‑proof direction to the true state we care about. |
>
> **Key idea** : we *peek* at the place where the model is most uncertain, borrow a direction that corrects that uncertainty, and use it to update the current sample.
>
>
> 2. **Why this look‑ahead step is robust - from robust optimization perspective**
>
> $$
> \min_x f_t(x + \delta),\
> \text{ where } \
> \delta = \arg\max_{\substack{\delta' \in \mathcal M}}
> \hat f_t(x, \delta').
> $$
>
> * **Inner maximization ( $\delta(x)$ )**
>   Chooses the perturbation inside the admissible set $\mathcal M$ that makes the surrogate loss $\hat f_t$ worst.
>   ➜ *Find the scenario where the current score estimate is least trustworthy.*
>
> * **Outer minimization**
>   Once that worst case is fixed, we ask: *“What move of $x$ will reduce that worst loss the most?”*
>   The answer is to step along the **negative gradient at the worst point** (our $d$).  This follows standard convex duality: the steepest descent direction minimises the first‑order approximation of $f_t$ under that fixed worst perturbation.
>
> * **Why this is safe**
>   Because the score field is Lipschitz‑continuous, every other perturbation in $\mathcal M$ produces a loss that is **no worse** than the one at $\delta(x)$.  Therefore, a direction that decreases the loss at the worst perturbation will decrease (or at least not increase) the loss for *all* admissible perturbations, i.e., the surrounding uncertainties.
>
>
> 3. **Conditions for the guarantee**
> - **Smooth nearby score** – The score network is locally Lipschitz (smooth in a small neighborhood).
> - **Error is bounded** – We assume the network’s score is never “too far off” from the true score inside the small ball we inspect.
> - **Step stays small** – The time step $\Delta t$ is small enough for the second‑order Taylor expansion to remain valid.
>
>
> Under these three assumptions the look‑ahead worst‑case update provably minimises the one‑step worst‑case loss and, as shown in Appendix A, yields more robust sampling in practice.
>
> ---
>
> ### **Q3: Clarification on Algorithm 2**
> Thanks for asking. The standard diffusion sampler solves  $\min_x\ f_t(x).$
> RODS instead solves the following **bilevel** problem (introduced right after line 232), which is the core of Algorithm 2:
> $$
> \min_x f_t(x + \delta),\
> \text{ where } \
> \delta = \arg\max_{\substack{\delta' \in \mathcal M}}
> \hat f_t(x, \delta').
> $$
>
> Two cases we addressed in the paper are:
>
> 1. **Sharpness‑Aware Sampling (SAS)** (lines 219–223):  $\hat f_{t,\delta'}(x,\delta') = f_t\bigl(x + \delta'\bigr).$
> 2. **Curvature‑Aware Sampling (CAS)** (line 224):  $\hat f_{t,\delta'}(x,\delta') = \bigl ||\nabla f_t(x + \delta')\bigr ||.$
>
> Algorithm 2 simply plugs in the chosen $\hat f_{t,\delta'}$ once the curvature detector $H(x_t)\geq\epsilon$ signals a “high‑risk” step, and then takes the corresponding robust update.
>
> ---
>
> ### **Q4: Evaluation on image quality**
>
> Thank you for encouraging a broader quality assessment. We have therefore computed standard image-quality and diversity metrics on 1,000 unconditional samples from FFHQ, AFHQv2, and 11k-hands. The table below reports FID, Inception Variance [1] (which quantifies variability in the latent representations), and DreamSim Diversity [2] (which measures the variance of features extracted from a batch of generated images, and our batch size is 64).
>
>
> | Dataset  | FFHQ                |        |         | AFHQ2               |        |         | 11k-hands        |        |         |
> |----------|---------------------|--------|---------|----------------------|--------|---------|-------------------------|--------|---------|
> |          | FID ↓               | Inception-Var ↑ | DreamSim ↑ | FID ↓               | Inception-Var ↑ | DreamSim ↑ | FID ↓               | Inception-Var ↑ | DreamSim ↑ |
> | EDM-Euler    | 20.48              | 0.0669 | 0.407   | 13.36              | 0.0539 | 0.486   | 14.57          | 0.0327                    | 0.1427|
> | EDM-Heun      | 19.56          | 0.0699 | 0.422 | 12.66          | 0.0548 | 0.488 | 14.20          | 0.0332                    | 0.1476               |
> | RODS-CAS      | 22.73              | 0.0661 | 0.400   | 16.08              | 0.0542 | 0.481   | 16.68          | 0.0318                    | 0.1509               |
>
> These results show that RODS-CAS preserves diversity while keeping FID comparable. The modest FID rise is largely explained by background smoothing: our curvature test sometimes removes ambiguous, high-frequency textures in the complex backgrounds, which the FID metric interprets as loss of detail. However, such effect could be analogous to portrait-mode photography, where a deliberately blurred background can worse a texture-based metric (FID) without reducing perceived quality. In our generation,  the main subjects remain sharp and artifact-free, which is crucial as it is exactly the goal of our correction. **_This is further supported by the 11k-hands experiment, where our method improves hand confidence (HAND-CONF) from 0.9216 (EDM-Euler) and 0.9227 (EDM-Heun) to 0.9347 (RODS-CAS)._**  HAND-CONF is computed by running a pre-trained hand detector [3] and calculating the confidence scores of the detections. This metric is widely used to assess the anatomical plausibility of generated hands. Because hands are structurally well-defined and challenging to synthesize, this benchmark provides a sensitive and interpretable proxy for visual realism. Overall, **_RODS-CAS delivers equally diverse, comparably high-quality unconditional samples while reducing hallucinations._**
>
> [1] Miao, Zichen, et al. "Training diffusion models towards diverse image generation with reinforcement learning." CVPR. 2024.
> [2] Domingo-Enrich, Carles, et al. "Adjoint matching: Fine-tuning flow and diffusion generative models with memoryless stochastic optimal control." arXiv preprint 2024.
> [3] Zhang, Fan, et al. "Mediapipe hands: On-device real-time hand tracking." arXiv preprint 2020.

---

> > ### Comment · Reviewer_ZPMe · 2025-08-06
> >
> > I would like to thank the authors for the detailed response, which clarifies the conceptual ideas behind the algorithmic design. The added experimental results address my main concern by showing that the proposed method preserves sampling quality. I will raise my score.

---

> > > ### Author Response · Authors · 2025-08-06
> > >
> > > Dear Reviewer ZPMe,
> > >
> > > Thank you so much for your thoughtful comments and for taking the time to engage with our rebuttal. We're truly glad to hear that our clarifications and additional results addressed your concerns, and we appreciate your willingness to update your score. Your feedback has been very encouraging for us.

---

### Official Review · Reviewer_jSsH · 2025-07-03

**Clarity:** 3
**Significance:** 2
**Originality:** 2
**Rating:** 5
**Confidence:** 3

**Summary:**

This paper tackles the hallucination problem in diffusion models by viewing diffusion sampling as a optimization procedure, which frames each sampling steps as a proximal sub-problem on a gradually sharpening potential landscape.
Based on this, the authors introduce Robust Optimization-inspired Diffusion Sampler, RODS, a novel curvature-based sampler detects sudden changes in score vector field, which can auto-adjust sampling directions and thus improve robustness and remove hallucination.
Importantly, RODS does not introduce any modification to the pretrained diffusion.
Experiments on AFHQ-v2 (cats/dogs/wild), FFHQ (faces) and 11k-hands show that RODS-CAS detects ≥ 70 % of annotated hallucinations and corrects > 25 % of faulty samples.

**Questions:**

- Why using curvature metric to detect hallucination?
I don't see why fluctuation in score magnitude can infer hallucination. Is it possible that in some region, score magnitude can be a false alarm? Maybe some region of the image has this type of score behavior.
- Can you show some image generation quality evaluations? For example, FID, NLL. Just want to see that this sampler doesn't tread hallucination to generation quality.
- How you make sure that the human labeled data is trustable?
- For hyperparameter epsilon and rho, can you provide a principled way to choose?

**Ethical Concerns:**

["NO or VERY MINOR ethics concerns only"]

**Final Justification:**

My main questions and concerns are answered. Therefore I recommend to accept the paper.

**Limitations:**

yes

**Quality:**

2

**Strengths And Weaknesses:**

Strengths:
- Paper is well structured and easy to follow, also algorithms and theorems / proofs are self-contained:
Nice derivations bridge Euler updates to the proximal optimization, and novel link between robust optimization and diffusion sampling. Unlike the previous prior temporal-variance filters, the Curvature-driven risk detector focus more on the shape of the score landscape,
- Comprehensive experiments across three datasets with manual hallucination annotation.
- Nearly no extra cost: since the method is plug-and-play, there is no change to the model.

Weaknesses:
- Although promising experimental results, the authors only provide hallucination related metrics. I think it will be more reasonable to also show some image quality evaluations, e.g. NLL, FID -- just want to make sure that while solving the hallucination issues, the image quality and density estimate are also guaranteed to be good.
- It's nice to use curvature index to reflect fluctuation in score magnitude, however, making connection between curvature and hallucinate is not rigorously illustrated. To me this fluctuation in score magnitude just tells me how rapid the score changes in this area, and maybe I should adjust my step size to be smaller to avoid introducing more error. These may be just correlation not causality.

---

> ### Author Rebuttal · Authors · 2025-07-31
>
> ### **W1&Q2: Evaluation on image quality**
> Thank you for encouraging a broader quality assessment. We have therefore computed standard image-quality and diversity metrics on 1,000 unconditional samples from FFHQ, AFHQv2, and 11k-hands. The table below reports FID, Inception Variance [1] (which quantifies variability in the latent representations), and DreamSim Diversity [2] (which measures the variance of features extracted from a batch of generated images, and our batch size is 64). Because RODS adjusts only the sampling trajectory and never alters the trained score network, its negative log-likelihood (NLL) is identical to that of the baseline sampler and is omitted for brevity.
>
> | Dataset  | FFHQ                |        |         | AFHQ2               |        |         | 11k-hands        |        |         |
> |----------|---------------------|--------|---------|----------------------|--------|---------|-------------------------|--------|---------|
> |          | FID ↓               | Inception-Var ↑ | DreamSim ↑ | FID ↓               | Inception-Var ↑ | DreamSim ↑ | FID ↓               | Inception-Var ↑ | DreamSim ↑ |
> | EDM-Euler    | 20.48              | 0.0669 | 0.407   | 13.36              | 0.0539 | 0.486   | 14.57          | 0.0327                    | 0.1427|
> | EDM-Heun      | 19.56          | 0.0699 | 0.422 | 12.66          | 0.0548 | 0.488 | 14.20          | 0.0332                    | 0.1476               |
> | RODS-CAS      | 22.73              | 0.0661 | 0.400   | 16.08              | 0.0542 | 0.481   | 16.68          | 0.0318                    | 0.1509               |
>
> These results show that RODS-CAS preserves diversity while keeping FID comparable. The modest FID rise is largely explained by background smoothing: our curvature test sometimes removes ambiguous, high-frequency textures in the complex backgrounds, which the FID metric interprets as loss of detail. However, such effect could be analogous to portrait-mode photography, where a deliberately blurred background can worse a texture-based metric (FID) without reducing perceived quality. In our generation,  the main subjects remain sharp and artifact-free, which is crucial as it is exactly the goal of our correction. **_This is further supported by the 11k-hands experiment, where our method improves hand confidence (HAND-CONF) from 0.9216 (EDM-Euler) and 0.9227 (EDM-Heun) to 0.9347 (RODS-CAS)._**  HAND-CONF is computed by running a pre-trained hand detector [3] and calculating the confidence scores of the detections. This metric is widely used to assess the anatomical plausibility of generated hands. Because hands are structurally well-defined and challenging to synthesize, this benchmark provides a sensitive and interpretable proxy for visual realism. Overall, **_RODS-CAS delivers equally diverse, comparably high-quality unconditional samples while reducing hallucinations._**
>
> Additional conditional metrics such as CLIP-score are not applicable here, as we work in the unconditional setting. We will incorporate this table and the accompanying discussion into the revised manuscript to present a more complete picture of image quality alongside hallucination reduction.
>
>
> [1] Miao, Zichen, et al. "Training diffusion models towards diverse image generation with reinforcement learning." CVPR. 2024.
> [2] Domingo-Enrich, Carles, et al. "Adjoint matching: Fine-tuning flow and diffusion generative models with memoryless stochastic optimal control." arXiv preprint 2024.
> [3] Zhang, Fan, et al. "Mediapipe hands: On-device real-time hand tracking." arXiv preprint 2020.
>
> ---
>
> ### **W2&Q1: Connection between curvature, uncertainty, and hallucination**
> Thank you for asking us to clarify the connection between curvature, uncertainty, and hallucination. The key hypothesis is that **hallucination correlates with model uncertainty** [1], which can be quantitatively captured by the **“rapid fluctuations (curvature)”** of the density field. Experimentally, we showed that the hallucination is not caused by various step size (Figures 9–10), but rather a directional error within low-density, high-uncertainty regions: merely changing the step-size of the sampler does not help if the direction itself is unreliable.
>
>
> **Curvature is a proxy for model uncertainty**
>
> - **Score field.**  We denote the learned score by $v(x) = \nabla_x \log P_t(x)$.
> - **Curvature index.**  As stated in §4.1 (Eq. 8), we define our curvature index is:
>
>
> $$
> H(x) = \|\nabla_x \|v(x + \delta)\| - \nabla_x \|v(x)\|\|,
> \quad
> \delta = \arg \max_{\|\delta\| = \rho} \|v(x + \delta)\|,
> $$
>
>
> Because $H(x)$ compares the gradient of $\|v\|$ at two nearby points, it measures **second-order variation** of the density field, i.e., how smoothly the score field changes in space. This reflects the key idea that **hallucination correlates with model uncertainty** [1].  By probing these rapid, spatial fluctuations, i.e., second‑order changes or “curvature”, we obtain a quantitative signal for potential sampling failures. In Appendix A.2, we further describe the conceptual foundation in detail:
>
> > “In well-behaved regions, this vector field changes smoothly. But in low-density areas, $v(x)$ can shift direction quickly—these are the regions where hallucinations are likely to occur.” (line 467–471)
>
> > “Rapid changes in this quantity often indicate instability in the local geometry of the score function.” (line 475–476)
>
>
> **Why “just shrink the step” is insufficient**
>
> Intuitively, shrinking the step size is a natural approach. We evaluated a naive fix, using a much smaller step size by running an EDM-Euler sampler with 200 uniform steps (Figures 9–10). Although each step is five times shorter than the 40-step baseline, hallucinations still appear. In contrast, in tons of our experiments, we often observed that RODS-CAS actively changes the update direction whenever the curvature index is high and succeeds in removing most of these artifacts. This comparison shows that the problem lies in directional error within low-density, high-uncertainty regions: merely changing the step-size of the sampler does not help if the direction itself is unreliable. Curvature information is therefore essential for steering the trajectory away from such regions.
>
> [1] Sumukh K. Aithal, Pratyush Maini, Zachary Lipton, and J. Zico Kolter. *Understanding hallucinations in diffusion models through mode interpolation.* Neurips 2024.
>
>
> ---
>
> ### **Q3: Human Evaluation Trustworthiness**
>
> As shown in Appendix B.6, our rating rubric is detailed.
>
> > *Face images (AFHQv2, FFHQ)* hallucinations are identified and annotated based on human perception, common sense, and visual plausibility. *Hand images (11k-hands)* are each assigned to one of seven categories: normal; extra fingers; missing fingers; incorrect finger structure; abnormal palm; multiple hands; unrecognizable/implausible.
>
> All images were labeled independently by two human raters to ensure objectivity. Whenever their judgments diverged, the pair reviewed the cases together and discussed until they reached a consensus, producing a single, agreed-upon label for each image. This consensus process both minimized individual bias and improved the overall reliability of our ground-truth data.
>
> To evaluate RODS-CAS’s impact relative to the baseline (EDM-Euler), annotators first identified hallucinations in the Euler-sampled outputs. They then compared each corrected image against its Euler counterpart, categorizing the result as “better,” “worse,” “same," or "unclear". This side-by-side comparison made it easy to see whether our method truly reduced artifacts, or introduced new ones. We will further update our Appendix B.6 with more details.
>
> Finally, in the interest of transparency and reproducibility, we will make our complete annotation guidelines, raw labels, and final consensus labels publicly available. This will allow other researchers to verify our results and build on our work with full confidence in the human-evaluation process.
>
> ---
>
> ### **Q4: Selection of Hyperparameter**
>
> Thank you for the question. Our paper has provided simple, data-driven principles for both hyper-parameters, detailed as follows. We will study the automated hyperparameter selection in the future.
>
>
> **$\epsilon$ — detection threshold (Section 4.1, Appendix B.2)**
>
> * **Role:** controls sensitivity; lower ε is more sensitive to the score changing, i.e., uncertain steps.
> * **Guideline (line 205–207):** “In high-stakes domains such as medical imaging, a lower $\epsilon$ may be preferred to prioritise sensitivity.”
>
> **$\rho$ — perturbation radius (Section 4.1)**
>
> * **Role:** sets the neighbourhood size over which we probe the score field.
> * **Choice:** match $\rho$ to the characteristic feature scale of the dataset. If the surrounding region is trustworthy (e.g., low visual complexity), a larger $\rho$ is safe; if not, a smaller $\rho$ focuses detection on truly uncertain zones.

---

> > ### Comment · Reviewer_jSsH · 2025-08-09
> >
> > Thank you for responding in details. My concerns are resolved, therefore I'm willing to increase my score to accept.

---

> ### Author Response · Authors · 2025-08-06
>
> Dear Reviewer jSsH,
>
> Thank you for taking the time to review our paper and providing your insightful feedback. We also appreciate you sharing the mandatory acknowledgement.
>
> As the author–reviewer discussion period concludes in two days, we would be grateful if you could share your thoughts on our rebuttal at your earliest convenience. If you have any remaining questions or concerns, we’d be happy to engage in further discussion. If our responses have resolved your concerns, we would greatly appreciate your reconsideration of our score.

---

### Official Review · Reviewer_GnZW · 2025-07-03

**Clarity:** 3
**Significance:** 3
**Originality:** 3
**Rating:** 5
**Confidence:** 4

**Summary:**

The paper proposes RODS, a novel plug-and-play method for mitigating hallucinations in diffusion models, which is an under-explored field of research. The authors interpret diffusion sampling as a continuation method in numerical optimization and integrate robust optimization principles to dynamically detect and correct "high-risk/unstable" steps in the sampling process, thereby alleviating hallucinations. RODS considers curvature changes as a proxy for detecting hallucinations, and proposes two updating strategies (SAS and CAS). The authors evaluate RODS on AFHQv2, FFHQ, and Hands datasets, demonstrating promising results for hallucination mitigation.

**Questions:**

- Is it possible to make \epsilon not specific to a particular dataset? or to derive epsilon in a data-independent manner? This would help the generalization of the proposed algorithm, especially in cases of high diversity datasets
- Please include the details of the survey and the qualitative metrics used
- Discuss the limitations in detail along with the side-effects of corrections such as semantic drift/ over-sanitization
- As pointed out in the weaknesses and suggestions section, I believe the paper is lacking an evaluation/analysis on the potential trade-off between generative capability and hallucination mitigation.

**Ethical Concerns:**

["NO or VERY MINOR ethics concerns only"]

**Final Justification:**

The paper proposes a grounded solution towards mitigating hallucinations in diffusion models, which is an under-explored area. The authors were able to coherently address most of the weaknesses and concerns raised while proposing edits to incorporate them in their final revision. Contingent on the authors including the suggested edits which makes the paper much stronger, I vote towards accepting this work --- raising my score to Accept from Borderline Accept.

**Quality:**

2

**Strengths And Weaknesses:**

**Strengths**
- The interpretation of the diffusion process as a continuation method, followed by the adaptation of robust optimization as a mitigation strategy is an elegant and theoretical sound approach.
- The paper addresses an underexplored research avenue and demonstrates strong hallucination detection rates and mitigation
- The proposed work does not require retraining (plug-and-play) with a relatively lower overhead in inference time
- Includes a detailed set of ablations
- a clear and well-written paper where the proposed algorithm is clearly motivated

**Weaknesses and Suggestions**
- The proposed method may suppress the generative ability of a diffusion model similar to mode collapse. While this would be acceptable to a certain degree given the hallucinations are mitigated, I believe an analysis of potential trade-offs between hallucination and generation diversity is required.
- While the use of the sharp curvature as a proxy is justifiable, I would suggest the authors discuss the underlying assumptions behind it as there could be naturally sharp but "valid" features
- The authors use the qualitative analysis results as one of their main metrics but do not present details of the survey. This would be needed in making sure the survey best represents the general community and reproducibility. Further, the use of hand-wavy ratings of terms like  "better", "worse", "same", and "unclear" needs to be explicitly defined (i.e., what qualifies for each category)
- The correction-induced changes by RODS may be unintended, where in certain cases a semantic drift and new subtle hallucinations are introduced by the correction process. For example in Fig. 9:
    - Row 1 left: Pose and complexion change.
    - Row 1 right: While the extra finger is removed, the pinky becomes implausibly long.
    - Row 3 left: Floating hand is not resolved.
    - Row 3 right: Newly introduced finger has inconsistent tone.
    - Row 4 right: Missing sleeve and changed hand orientation.
- The authors do not discuss on limitations, any architectural dependence, applicability to high-diveristy datasets where the notion of hallucination is itself ill-defined.
- The following additional related work appears to be in the same domain of mitigating hallucinations in unconditional diffusion models: *Oorloff et al.  "Mitigating Hallucinations in Diffusion Models through Adaptive Attention Modulation"*

---

> ### Author Rebuttal · Authors · 2025-07-31
>
> ### **W1 & Q4: Trade‑off between hallucination mitigation and generation diversity**
> We fully agree that a mitigation technique should *not* come at the cost of collapsing the model’s creative space. Intuitively, **RODS touches only a small subset of trajectories, so most images remain exactly the same**. Visual evidence of this can be found in Appendix B.7 (Figures 13 and 14), while standard diversity metrics further confirm this observation.
>
> Below we quantify and explain why RODS does *not* suppress diversity. In particular, we assess diversity across 1k samples using: **Inception Variance** [1], which quantifies variability in the latent representations, and **DreamSim Diversity** [2], which measures the variance of features extracted from a batch of generated images (batch size = 64). The experiment results show that RODS-CAS preserves generation diversity on par with the EDM-Euler method while effectively mitigating hallucinations, as detailed in the paper.
>
> | Dataset| FFHQ |   | AFHQ2    |  | 11k-hands | |
> |----------|---------------------|-------------------|----------------------|-------------------|----------------------|-------------------|
> |    | Inception-Var ↑     | DreamSim ↑        | Inception-Var ↑      | DreamSim ↑        |Inception-Var ↑      | DreamSim ↑        |
> | EDM-Euler    | 0.0669              | 0.407             | 0.0539               | 0.486             | 0.0327                    | 0.1427               |
> | EDM-Heun      | 0.0699          | 0.422         | 0.0548               | 0.488         | 0.0332                    | 0.1476               |
> | RODS-CAS | 0.0661              | 0.400             | 0.0542           | 0.481             | 0.0318                    | 0.1509               |
>
>
> [1] Miao, Zichen, et al. "Training diffusion models towards diverse image generation with reinforcement learning." CVPR. 2024.
> [2] Domingo-Enrich, Carles, et al. "Adjoint matching: Fine-tuning flow and diffusion generative models with memoryless stochastic optimal control." arXiv preprint 2024.
>
> ---
>
> ### **W2: Clarifying the assumptions behind the curvature proxy**
>
> We appreciate the reviewer’s comment on discussing the assumptions underlying our curvature-based detection. The key hypothesis is that **hallucination correlates with model uncertainty** [1]. As mentioned in our paper, "in well-behaved regions, this vector field changes smoothly. But in low-density areas, $v(x)$ can shift direction quickly—these are the regions where hallucinations are likely to occur" (line 467-471). By probing these rapid, spatial fluctuations, i.e., second‑order changes or “curvature”, we obtain a quantitative signal (our curvature index $H(x)$) for potential sampling failures.
>
> In terms of **the naturally sharp but valid features,** such structures (e.g., object edges) are **unrelated** to the curvature index $H(x)$ used in our method.  While such sharpness in image exhibits high **image gradients** $\nabla_x I(x)$, this differs fundamentally from the **density-field gradient** $v(x) = \nabla_x \log p_t(x)$, which reflects changes in data density.  Our curvature index $H(x)$ measures variations in this score field—not in image intensity.
>
> [1] Sumukh K. Aithal, et al. *Understanding hallucinations in diffusion models through mode interpolation.* Neurips 2024.
>
> ---
>
> ### **W3 & Q2: Detailing our qualitative survey and rating rubric**
> We thank the reviewer for pointing this out.
>
> **For hallucination labeling (Appendix B.6) :**
>
> -> *Face images (AFHQv2, FFHQ)* hallucinations are identified and annotated based on human perception, common sense, and visual plausibility. *Hand images (11k-hands)* are each assigned to one of seven categories: normal; extra fingers; missing fingers; incorrect finger structure; abnormal palm; multiple hands; unrecognizable/implausible.
>
> **Current use of “better”/“worse”/"same", and "unclear".**
>
> To make our evaluation protocol more precise, we conducted pairwise comparisons by **human raters on anonymized image pairs** (RODS vs. baseline). Unlike traditional binary hallucination labels, these judgments reflect relative image quality rather than the mere presence or absence of hallucinations. As a result, it provides a more nuanced and reliable assessment of generation fidelity. **_Furthermore, in the interest of transparency and reproducibility we will share our code, data, and labels in Github._** More specifically, the detailed rubric is shown below:
>
> - **Better**: The RODS-generated image is perceived as more faithful or realistic, often due to improved correction of visual anomalies (e.g., fixing a misplaced eye) without introducing new defects.
> - **Worse**: The RODS-generated image is clearly degraded relative to the baseline, typically because it introduces new artifacts or hallucinated elements that were not present in the original.
> - **Same:** No human-identifiable difference exists between the baseline and the RODS-generated image.
> - **Unclear:** A change occurs, but it does not involve new hallucinations, and it is difficult to judge which version is better (e.g., slight pose changes or missing accessories like a ring).
>
> ---
>
> ### **W4 & Q3: Side-effects of corrections**
>
> We really appreciate the reviewer’s careful analysis and thoughtful comments—these are valuable and nuanced issues that touch on the core difficulty of hallucination correction. Due to the inherently ambiguous definition of hallucination, our correction strategy is not able to make fully “intended” edits. Rather, RODS introduces conservative updates at points identified as high-risk under the base model’s score landscape.
>
> 1. **Imperfect correction**: The effectiveness of correction is fundamentally limited by the base model itself. For instance, in cases like Row 3 (right), where the baseline model tends to denoise the fifth finger into blank space, our robust update may reintroduce it in a subtle but imperfect form (e.g., slightly discolored or ill-shaped), as we can only perturb within the model’s generative prior.
>
> 2. **Semantic drift**: On one hand, semantic drift is less relevant in our experiments, as there is no external prompt or class label whose meaning could shift. On the other hand, different diffusion timesteps correspond to different levels of abstraction. Among changes, edits made at early stages (high t) can result in semantic drift, while those at later stages (low t) typically affect finer details. Our current method does not constrain which timesteps are eligible for correction, as we found that some large-scale hallucinations require intervention at earlier stages. However, such early-stage perturbations may also affect high-level semantics, such as altering the orientation of a hand, as seen in Row 4 (right). If semantic consistency is a priority, one could optionally restrict updates to later timesteps to avoid undesired high-level changes.
>
> 3. **Over-sanitization**: Quantitative diversity remains almost the same: as shown in W1&Q4. Additionally, since most samples are left bit-wise unchanged (Fig 13-14), the model’s creative range is effectively preserved, addressing the over-sanitization concern. We will post all 1k results and the human label for public review.
>
>
> We acknowledge that achieving more targeted and controllable corrections would require more precise hallucination localization and timestep-aware update scheduling. We view this as a promising direction for future work.
>
> ---
>
> ### **W5: More discussion on limitation**
>
> Thank you for highlighting these important considerations. Here is how we will address them in the revised manuscript:
>
> 1. **Architectural dependence (VE formulation)**
> Our RODS algorithm can be directly implemented within the VE-ODE framework, which benefits from the equivalence between VE-based sampling and the continuation method under the usual parameterization (Section 3). We will clarify this dependency and note that extending RODS to other sampling schemes (e.g., VP or latent-space diffusion) is conceptually straightforward but remains to be validated experimentally.
>
> 2. **Dataset scope limitations**
> Currently, we have tested RODS on well-established benchmarks (AFHQv2, FFHQ, 11k-hands), where hallucinations can be easily observed without domain expertise. We acknowledge that in highly diverse or specialized domains, i.e., medical imaging, the definition of “hallucination” must be grounded in expert judgment. We are collaborating with domain practitioners to evaluate RODS under these conditions.
>
> ---
>
> ### **W6: Relevant work suggestion**
>
> Thank you for pointing out this relevant work. While it takes a different approach (adaptive attention modulation) compared to our trajectory-level geometric regularization, we agree that it addresses a related goal. As it was posted on arXiv in February 2025 and had no available code at the time of our submission, a direct experimental comparison was not feasible. We will gladly include a citation and discuss its relation to our work in the revised section.
>
> ---
>
>
> ### **Q1: Selection of epsilon**
>
> In our current experiments, we observed that the choice of $\epsilon$ is influenced by both the **dataset characteristics** and the **desired sensitivity level**. For instance, 11k-hands features clean backgrounds and fine-grained hand details, so a smaller $\epsilon$ helps detect subtle hallucinations in localized regions. In contrast, face datasets (AFHQv2, FFHQ) often involve more complex backgrounds, where a slightly larger $\epsilon$ is preferred to avoid overreacting to natural variation in curvature. More discussion about the epsilon selection has been presented in Section 4.1 (line 205-207) and Appendix B.2. Given a desired sensitivity level, we believe it is possible to automatically choose $\epsilon$ (there are similar approaches in traditional literature), but we consider it as a future research direction.

---

> ### Author Response · Authors · 2025-08-06
>
> Dear Reviewer GnZW,
>
> Thank you once again for your time and valuable feedback on our paper. We have carefully considered your detailed comments and addressed your concerns in our rebuttal above.
>
> If you have any further questions or would like us to clarify any points, we would be more than happy to provide additional details or engage in further discussion. If you feel our response addresses your concerns, we would appreciate your reconsideration of our score.

---

> ### Comment · Reviewer_GnZW · 2025-08-06
> **Follow-up to the rebuttal by authors**
>
> I appreciate the authors' efforts to respond to the raised concerns, where most of my concerns have been addressed. However, I have a few follow-up questions for the authors.
>
> ---
> ### W1 & Q4: Trade‑off between hallucination mitigation and generation diversity:
> Including DreamSim and Inception Variance helps in quantifying the generative capability, where we see that the proposed model reports slightly weaker metrics compared to the other models except for DreamSim in 11K Hands --- but yet may be considered at par as the authors have stated. In addition to the reported comparisons, I believe the authors need to include the metrics for the default diffusion models (w/o any mitigation strategy) to set the baseline for the generative capability.
>
> ---
> ### W4 & Q3: Side-effects of corrections
>
> I agree with the stance of the authors, where mitigating hallucinations at early denoising steps could lead to unwarranted/uncontrollable side effects such as semantic drift. While it is true that all the results shown are in unconditional settings, consider the practical aspect of using RODS with the generation.
>
> For example, if a user simply wants a bunch of random hands, they could either filter out the hallucinations (using a detector as proposed in [1]) or alternatively use RODS to create a pool of hand images (which would be similar to the filtered sample set). I would appreciate if the authors could explain why would once choose one over the other.
> Additionally, while I don't undermine the authors' work by any means, I believe that correcting a hallucination artifact while preserving the other semantics of the image has a much higher practicality. For example a user would be interested in correcting a hallucination of a particular image because the rest of the semantics are appealing/of use, else would perhaps choose to discard (given the model is capable of generating a large corpus).
>
> ---
> ### W5: More discussion on limitation | 2. Dataset scope limitations
> I agree that a domain such as "medical imaging", could be handled through expert judgement. However, consider a diffusion model that is trained on a much diverse corpus (e.g. Stable Diffusion trained on LAION5B), in such cases I would like to know the authors take on the scalability of the proposed algorithm. From my understanding, the RODS is limited to diffusion models trained on datasets with one/few/less-diverse classes.
>
> ---
> Finally, I would like the authors to list the planned amendments to the final manuscript. The authors need not go into details, a concise list would suffice.
>
>
>
> [1] Sumukh K. Aithal, et al. Understanding hallucinations in diffusion models through mode interpolation. Neurips 2024.

---

> > ### Author Response · Authors · 2025-08-07
> > **Response to the Follow-up Questions by Reviewer GnZW**
> >
> > We sincerely appreciate the reviewer’s time and thoughtful feedback. Please find our detailed responses below.
> >
> > ---
> >
> > ### **FQ1: Baseline Performance of the Default Diffusion Model**
> >
> > Thanks for pointing out the unclear area. As we mentioned in the paper, RODS is a plug-and-play method. If $\epsilon$ (the detection threshold of RODS) is set sufficiently high, no hallucinations are flagged; in this limit, RODS reduces exactly to the default diffusion model. This default diffusion model is **EDM-Euler**, whose metrics have been reported. We will clearly note such equivalence in the revision.
> >
> > ---
> >
> >
> > ### **FQ2: More discussion on the side-effects and practicality of corrections**
> >
> > We thank the reviewer for raising this important point. If we understand correctly, your questions focus on: 1) Practically, when one might prefer a filter-and-discard approach versus RODS (on-the-fly correction); 2) In unconditional generation, how RODS might preserve appealing semantics while fixing local hallucinations.
> >
> > Below, we address each in turn:
> >
> > **1. Filter-and-Discard vs. RODS (Correction)**
> >
> > | Aspect          | Filter + Discard (\[1])                | RODS (Correction)                      |
> > | --------------- | -------------------------------------- | ------------------------------------------------- |
> > | **Efficiency**  | Efficient when hallucinations are rare, **more efficient** | More efficient when hallucinations are much more common ¹|
> > | **Handling Specific Artifacts** | Must discard entire sample if any hallucination occurs | Can patch only the problematic region, **more flexible**    |
> >
> > ¹ For example, if the hallucination rate is 75%, generating 100 good images via filtering requires 400 total samples. In contrast, RODS only needs to generate detected and corrected 100 samples.
> >
> >
> >
> >
> > **2. Preserving Semantics While Fixing Local Artifacts**
> >
> > We share the reviewer’s concern about preserving desirable semantics. In our framework, most of the image remains unchanged—only the images with localized regions identified by the $H(x)$ index are modified. As demonstrated in Figures 3–5, the corrections are generally small and targeted. If semantic consistency is a priority, one could optionally restrict updates to later timesteps (towards $t=0$) where the image is generating the local details.
> >
> >
> > **Please let us know if we understand and address your concern. We are always happy to engage in further discussion.**
> >
> >
> > [1] Sumukh K. Aithal, et al. Understanding hallucinations in diffusion models through mode interpolation. Neurips 2024.
> >
> > ---
> >
> >
> > ### **FQ3: Dataset Scope and Scalability**
> >
> > RODS is an adjusted diffusion sampling framework built on the hypothesis that hallucinations arise in regions of local uncertainty, which can be characterized by local geometric properties in the density field, i.e., the curvature or our proposed $H(x)$ index. **The method does not rely on any assumptions about the dataset structure itself**, making it generally applicable. While datasets with highly diverse distributions may influence the geometry of the density field, in our humble opinion, such effects could be addressed by **tuning RODS hyperparameters such as the detection or correction radius**. That said, validating this adaptability requires further empirical study, and we are eager to systematically evaluate RODS across a broader range of datasets to assess its robustness and generalization in practice.
> >
> > ---
> >
> >
> > ### **FQ4: Final Manuscript Checklist**
> >
> > 1. **Release Materials Publicly**: Publish all source code, generated image samples, and the hallucination evaluation form online for public access and reproducibility.
> > 2. **Manuscript Revisions**:
> >    - Include quantitative image generation metrics in the main text to illustrate output quality.
> >    - Provide more intuitive explanations about the connection between uncertainty, curvature, and hallucination.
> >    - Provide detailed descriptions of the qualitative survey process and the rating rubric in the appendix for transparency and replicability.
> >    - Expand the discussion section to cover observed side effects and limitations of the method.
> >    - Incorporate additional related literature as suggested by the reviewers.
> >
> > ---
> >
> > Once again, thank you for your valuable insights. Please let us know if our responses have addressed your concerns. We’re always happy to continue the discussion.

---

### Decision · Program_Chairs · 2025-09-17

**Decision:**

Accept (poster)

**Comment:**

* Summary
The paper tackles the problem of reducing hallucinations in generative models in a novel manner.
They recast the problem of hallucinations as an outlier and discontinuity selection problem, which can be solved with tools and methods from optimization theory and robust optimization. Practically, this is done by looking at the stochastic process of diffusion, recasting the dynamical equation in a dynamical programming style optimization procedure, and showing it's equivalent to robust optimization. Then show the methodology on several well-known datasets.

* Strengths/ reasons for acceptance
The paper is very well written and clear; the authors state all of their claims in a very crisp manner. They also justify it with the analytical definitions and some theoretical bounds. The empirical results on several different data sets also support the authors' claim. The code for the paper will be released and can be thoroughly tested.
From a scientific point of view, the paper is excellent, casting the problem into one from optimization theory, the results are even stronger since they can be regularly verified and have a massive body of literature with full proof results.
Authors also made an above-and-beyond effort to rebut and answer all of the questions the reviewers had and then some.
With all of the above, I think this is a very strong paper, and I would suggest accepting it

* Weakness
The first weak point of the paper is the lack of standardization of benchmarks in the field, which makes it slightly hard to compare and judge the results with other SOT.
The second issue is in how hallucinations are defined in the paper. It's a bit too loose, but then again, it's not just here; it's a general issue with the field.
The third weak point is indeed how the curvature of the loss function relates to hallucination and the optimization process, while the authors addressed this as part of the reviewers' GnZW questions, it's still a bit lacking in the justification part.

All of the above, though, do not reduce the novelty and significance of the paper, and I would suggest a strong acceptance